# MS-Diffusion: Multi-subject Zero-shot Image Personalization with Layout Guidance

**Xierui Wang[2,+]**   **Siming Fu[2,+]**   **Qihan Huang[2]**   **Wanggui He[1]**   **Hao Jiang[1,†]**

[1]Alibaba Group   [2]Zhejiang University
[+]Equal contribution   [†]Corresponding author

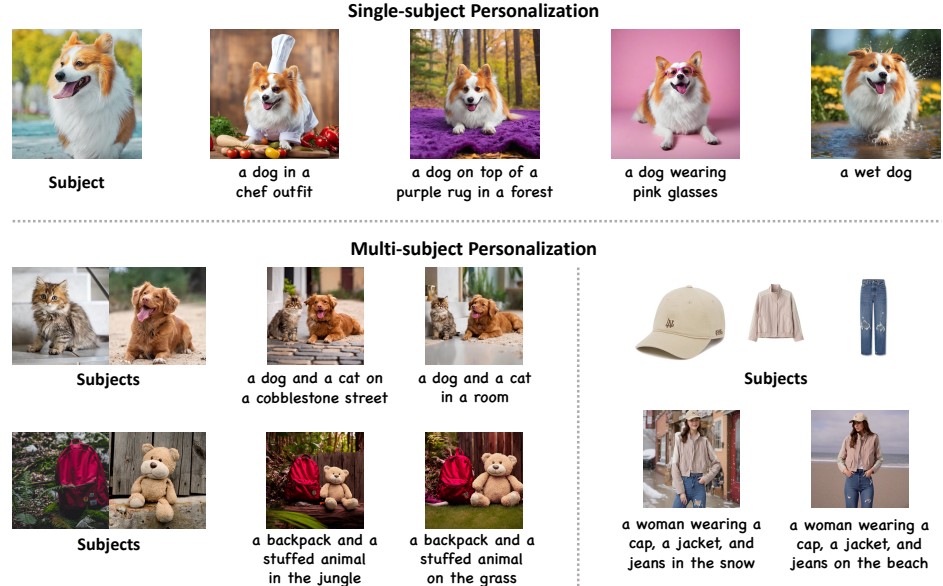

Figure 1: Representative outputs showcase the capabilities of MS-Diffusion in typical applications. The MS-Diffusion framework facilitates personalization across both single-subject scenarios (the upper panel) and multi-subject contexts (the lower panel). Notably, while preserving the intricacies of subject detail, MS-Diffusion achieves a marked enhancement in textual fidelity.

## Abstract

Recent advancements in text-to-image generation models have dramatically enhanced the generation of photorealistic images from textual prompts, leading to an increased interest in personalized text-to-image applications, particularly in multi-subject scenarios. However, these advances are hindered by two main challenges: firstly, the need to accurately maintain the details of each referenced subject in accordance with the textual descriptions; and secondly, the difficulty in achieving a cohesive representation of multiple subjects in a single image without introducing inconsistencies. To address these concerns, our research introduces the MS-Diffusion framework for layout-guided zero-shot image personalization with multi-subjects. This innovative approach integrates grounding tokens with the feature resampler to maintain detail fidelity among subjects. With the layout guidance, MS-Diffusion further improves the cross-attention to adapt to the multi-subject inputs, ensuring that each subject condition acts on specific areas. The proposed multi-subject cross-attention orchestrates harmonious inter-subject compositions while preserving the control of texts. Comprehensive quantitative and qualitative experiments affirm that this method surpasses existing models in both image and text fidelity, promoting the development of personalized text-to-image generation. The project page is `https://MS-Diffusion.github.io`.

# 1 INTRODUCTION

Recent advancements in text-to-image (T2I) diffusion methodologies (Rombach et al., 2022; Saharia et al., 2022; Betker et al., 2023) have propelled the field to new heights, realizing unprecedented levels of photorealism while demonstrating a refined ability to conform to textual prompts. These achievements have spurred the development of a broad spectrum of applications, most prominently in the domain of personalized T2I (P-T2I) models, which are tasked with the complex undertaking of assimilating and regenerating novel visual concepts or subjects across diverse contexts with a heightened demand for conceptual and compositional fidelity. Despite fine-tuning-based techniques such as DreamBooth (Ruiz et al., 2023) and Textual Inversion (Gal et al., 2023) yield results with considerable accuracy, they necessitate extensive resources for tuning individual instances and for the storage of multiple models, which renders them less feasible for widespread application. To circumvent these resource-intensive requirements, fine-tuning-free alternatives have come to the fore.

Single-subject driven personalization methods, IP-Adapter (Ye et al., 2023) and ELITE (Wei et al., 2023) for instance, introduce a specialized cross-attention mechanism that distinctly processes text and image features, thereby affording the possibility to employ reference images directly as visual prompts within the model. Furthermore, recent works have employed multi-subject driven customization methodologies to concatenate visual concepts with textual prompts, offering a glimpse of the potential in techniques like SSR-Encoder (Zhang et al., 2024), $\lambda$-ECLIPSE (Patel et al., 2024), Emu2 (Sun et al., 2023), and KOSMOS-G (Pan et al., 2023). These models harness identity data and amalgamate it with text via cross-attention, exhibiting proficiency in adjusting textures. Nevertheless, above zero-shot personalization methods encounter limitations, notably in adapting a pressing question pertains to the congruence of granular details between the subject depicted in the synthesized imagery and its corresponding subjects, along with the degree of semblance between the content of the generated image and associated textual descriptions. The challenge is further amplified in scenarios requiring the personalization of multiple subjects. Especially, the challenge of ensuring harmonious representation when multiple subjects are incorporated—specifically, the elucidation of whether the resultant image manifests any discordant elements or deleterious interactions in accordance with textual directives and multi-subject referential controls. As illustrated in Figure 2, multi-subject personalization methods frequently incur notable detail inaccuracies in a fine-tuning-free framework and often lead to subject neglect, subject overcontrol, and subject-subject conflict issue within the generated images.

To confront these identified challenges, we are the **first** to introduce the layout-guided zero-shot image personalization with multiple subjects (MS-Diffusion) framework, which consolidates the accommodation of multiple subjects, the incorporation of zero-shot learning capabilities, the provision of layout guidance, and the preservation of the foundational model's parameters. Firstly, we design the grounding resampler to extract the subject detailed features and integrate them with grounding information containing entities and boxes. As an image projection module, the proposed grounding resampler can enhance the subject fidelity while appending semantic and positional priors. Secondly, we propose a novel cross-attention mechanism for multiple subjects, which confines subjects to represent themselves in specific areas. This confluence not only facilitates the efficacious infusion of multi-subject data into the model but also mitigates conflicts between text and image subject control conditions. Such an approach culminates in the refined granularity of control over the image's multi-subject composition. The experimental results demonstrate our method consistently outperforms the state-of-the-art approaches on all the benchmarks. **We conclude the previous P-T2I works and provide an overall comparison in Table 1.** The contributions can be summarized as follows:

- We introduce a layout-guided, zero-shot multi-subject image personalization framework within the diffusion model paradigm, designated as 'MS-Diffusion'. This innovation streamlines the complex process of preserving detailed subject references. Moreover, it seamlessly integrates multiple subjects into a coherent and harmonious personalized image.
- The 'Grounding Resampler' is advanced as a novel feature refinement mechanism. This construct enriches the detail extraction from images by ascertaining the correlative content and fusing it with box embeddings that demarcate the anticipated spatial zones for each subject. Additionally, we introduce a specialized multi-subject cross-attention mechanism, confronting and rectifying prevalent complications in multi-subject personalization, including inadvertent subject neglect, disproportionate subject dominance, and internecine subject conflicts.

Table 1: **An overview of previous studies of P-T2I tasks.** MS-Diffusion is the **first** approach to support multi-reference zero-shot P-T2I generation with layout guidance and base model freezing.

| Method | Zero Shot | Multi Subject | Base Model Freezing | MLLM Free | Layout Guidance |
|---|---|---|---|---|---|
| Textual Inversion (Gal et al., 2023) | ✗ | ✗ | ✓ | ✓ | ✗ |
| DreamBooth (Ruiz et al., 2023) | ✗ | ✗ | ✗ | ✓ | ✗ |
| ELITE (Wei et al., 2023) | ✓ | ✗ | ✗ | ✓ | ✗ |
| BLIP-Diffusion (Li et al., 2023a) | ✓ | ✗ | ✗ | ✗ | ✗ |
| IP-Adapter (Ye et al., 2023) | ✓ | ✗ | ✓ | ✓ | ✗ |
| Emu2 (Sun et al., 2023) | ✓ | ✓ | ✗ | ✗ | ✗ |
| Kosmos-G (Pan et al., 2023) | ✓ | ✓ | ✓ | ✗ | ✗ |
| λ-ECLIPSE (Patel et al., 2024) | ✓ | ✓ | ✓ | ✗ | ✗ |
| SSR-Encoder (Zhang et al., 2024) | ✓ | ✓ | ✓ | ✓ | ✗ |
| **MS-Diffusion** | ✓ | ✓ | ✓ | ✓ | ✓ |

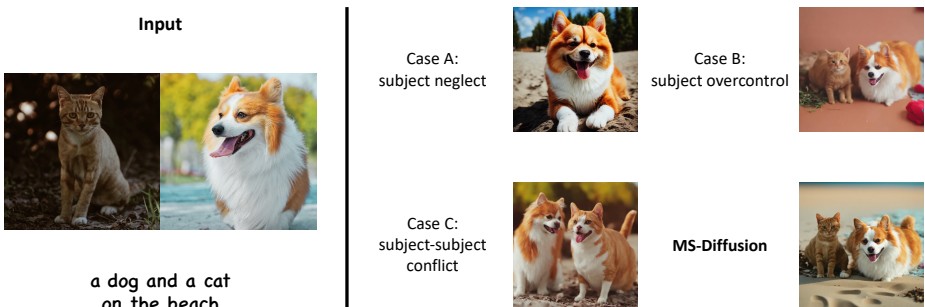

Figure 2: **Challenges inherent in multi-subject personalization approaches.** Through the explicit layout guidance, MS-Diffusion addresses these challenges by directing subject conditioning to specific areas, simultaneously maintaining high image fidelity.

- The capabilities of 'MS-Diffusion' are empirically substantiated through its ability to synthesize a broader spectrum of images with notable fidelity. The paper further delineates comprehensive ablation studies, underpinning the rationale behind our design decisions and affirming the efficacy of our proposed approach.

## 2 RELATED WORK

### 2.1 TEXT-TO-IMAGE GENERATION

Text-to-image generative models (Saharia et al., 2022; Bao et al., 2023; Esser et al., 2024; Podell et al., 2023) are capable of producing high-quality images using user-provided text prompts. In recent times, diffusion-based models have shown strong performance in text-to-image tasks. Stable Diffusion (Rombach et al., 2022) proposes conducting the diffusion process in latent space rather than pixel space, which reduces the sampling steps without compromising image quality. Kandinsky (Razzhigaev et al., 2023) takes both text embedding and image embedding as conditions to generate images more controllably. DALLE-3 (Betker et al., 2023) recaptions the training data pairs and utilizes T5-XXL (Chung et al., 2022) as the text encoder to strengthen the prompt-following ability. StableCascade (Pernias et al., 2023) presents a cascaded architecture to leverage outputs of front stages as priors, further reducing the latent space. PixArt-$\alpha$ (Chen et al., 2023a) also employs a large T5 text encoder and replaces the original U-Net backbone with a transformer (Peebles & Xie, 2023). These models focus on the basic text-to-image ability and cannot handle the situation when users provide specific subjects.

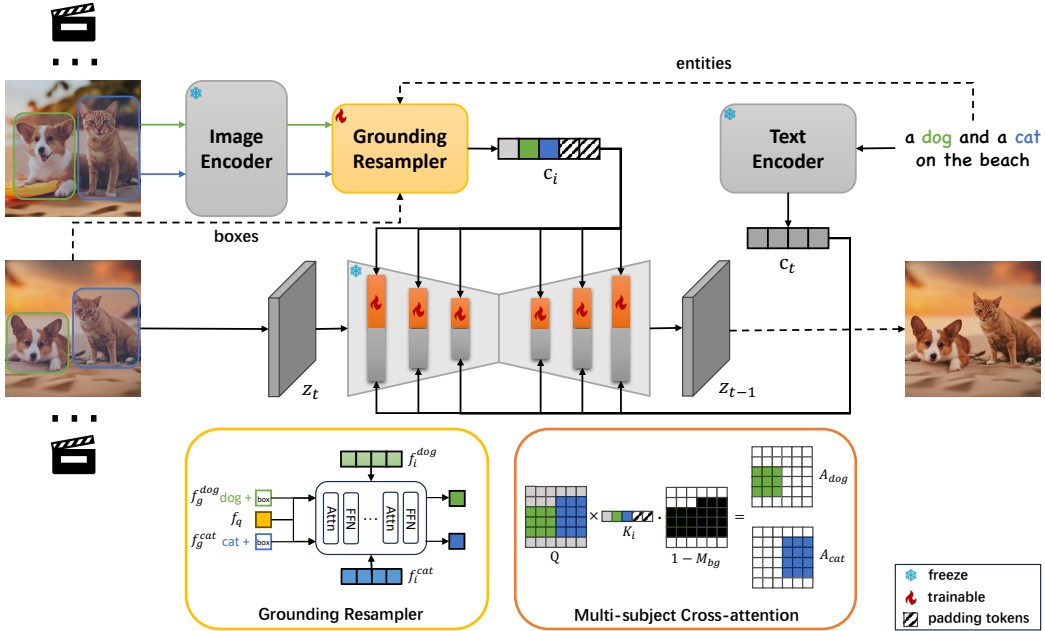

Figure 3: **The overall pipeline of MS-Diffusion.** It introduces two pivotal enhancements to the model: the grounding resampler and multi-subject cross-attention mechanisms. Firstly, the grounding resampler adeptly assimilates visual information, correlating it with specific entities and spatial constraints. Subsequently, the cross-attention mechanism facilitates precise interactions between the image condition and the diffusion latent within the multi-subject attention layers. Throughout the training phase, all components of the pre-existing diffusion model remain frozen.

To finely control text-to-image generation, some diffusion models support users in providing layout guidance. Layout Diffusion (Zheng et al., 2023) and GLIGEN (Li et al., 2023c) input positions and labels of bounding boxes into the diffusion model and train it to learn the layout information. DenseDiffusion (Kim et al., 2023) develops a training-free method and modulates the attention maps in the inference phase. Instance Diffusion (Wang et al., 2024b) and MIGC (Zhou et al., 2024) extend the layout-conditioned diffusion to the instance level, enabling the model to generate multiple objects with precise quantities. While layout-guided diffusion models have robust controllability, they cannot reference specific concepts, which is emphasized in personalized text-to-image generation.

## 2.2 TEXT-TO-IMAGE PERSONALIZATION

Text-to-image personalization (Han et al., 2023; Qiu et al., 2023; Chen et al., 2024; Shi et al., 2024; Hu et al., 2024) has attracted much attention in the community for its powerful referencing capability of both text and image prompts. Textual inversion (Gal et al., 2023) and DreamBooth (Ruiz et al., 2023) utilize an identifier in the text to bind the visual concept through fine-tuning. IP-Adapter (Ye et al., 2023) proposes a zero-shot personalized model by projecting the image embedding to the cross-attention layers. InstanceID (Wang et al., 2024a) develops an approach for identity personalization, replacing the image encoder with a face encoder and employing ControlNet (Zhang et al., 2023) to integrate the face landmarks. To narrow the gap between image and text prompts, Kosmos-G (Pan et al., 2023) and $\lambda$-ECLIPSE (Patel et al., 2024) conduct multi-modal training to unite the inputs by text-image interleaving. SSR-Encoder (Zhang et al., 2024) design a query network to extract a single subject from images with multiple subjects for personalization.

Though past research in this field has significantly enhanced the ability to reference single subjects, few studies have explored zero-shot multi-subject personalized models. Moreover, existing related works struggle to address conflicts in personalized generation with multiple subjects and generate bad results, which is precisely the focus of our work, to discuss and resolve these issues.

## 3   METHOD

The pipeline of MS-Diffusion is shown in Figure 3. Through an improved data construction strategy, we can get multiple subject images together with the corresponding entities and layouts as input. We propose a grounding resampler to separately extract the image features and integrate them with phrase embedding and box embedding for condition enhancement. Inside the cross-attention layers, we further introduce the masked cross-attention to guide the generation with layout priors and alleviate conflicts of multiple subjects. The training needs no pre-trained weights to be optimized and remains plug-and-play in various base models.

### 3.1   PRELIMINARIES

**Stable Diffusion with Image Prompt.**   As a widely used diffusion model, Stable Diffusion (SD) (Rombach et al., 2022) conducts the diffusion process in the latent space. Given an image and a text prompt, SD encodes them into latent code $\mathbf{z}$ and condition embedding $\mathbf{c_t}$ utilizing VAE (van den Oord et al., 2017) and CLIP (Radford et al., 2021) text encoder, respectively. In zero-shot image personalization architectures like IP-Adapter (Ye et al., 2023), images can also be considered a condition of the diffusion model. Specifically, a subject image is encoded to image embeddings by an image encoder and then projected into the original condition space of the diffusion model denoted as $\mathbf{c}_i$. For a timestep $t$ which is uniformly sampled from a fixed range, the model $\theta$ predicts the noise $\boldsymbol{\epsilon}_\theta$ and is optimized through the objective:

$$\mathcal{L}_{IP} = \mathbb{E}_{\mathbf{z},\mathbf{c},\epsilon,t}\left[\|\epsilon - \boldsymbol{\epsilon}_\theta\left(\mathbf{z}_t \mid \mathbf{c_t}, \mathbf{c_i}, t\right)\|_2^2\right] \tag{1}$$

where $\epsilon \sim \mathcal{N}(\mathbf{0}, \mathbf{I})$. In this work, we employ $\mathcal{L}_{IP}$ with SDXL (Podell et al., 2023) as the pre-trained model, which contains two CLIP text encoders and additional condition inputs besides $\mathbf{c}$ and $t$, omitted for brevity.

**Cross-attention.**   In IP-Adapter Ye et al. (2023), both $\mathbf{c}_i$ and $\mathbf{c}_t$ are integrated into the U-Net backbone through cross-attention layers:

$$\text{Attn}\left(\mathbf{Q}, \mathbf{K}_i, \mathbf{K}_t, \mathbf{V}_i, \mathbf{V}_t\right) = \gamma \cdot \mathbf{z}_{img} + \mathbf{z}_{txt} = \gamma \cdot \text{Softmax}\left(\frac{\mathbf{Q}\mathbf{K}_i^\top}{\sqrt{d}}\right)\mathbf{V}_i + \text{Softmax}\left(\frac{\mathbf{Q}\mathbf{K}_t^\top}{\sqrt{d}}\right)\mathbf{V}_t \tag{2}$$

where $\mathbf{Q} = \mathbf{z}W_q$, $\mathbf{K_i} = \mathbf{c}_i W_k^i$, $\mathbf{K_t} = \mathbf{c}_t W_k^t$, $\mathbf{V_i} = \mathbf{c}_i W_v^i$, $\mathbf{V_t} = \mathbf{c}_t W_v^t$, and $W_q$, $W_k$, $W_v$ are corresponding projection weight matrices, and $d$ represents the dimensionality of the key vectors. Note that the key and value matrix of $\mathbf{c}_i$ and $\mathbf{c}_t$ are independent of each other to decouple conditions of different modalities. Previous studies (Hertz et al., 2023; Tang et al., 2023) have found that attention maps $\mathbf{A} = \text{Softmax}\left(\mathbf{Q}\mathbf{K}^\top/\sqrt{d}\right)$ can reflect the attribution relation between generated images and conditions, which means that they determine the effect of condition controls.

### 3.2   DISCUSSION ON MULTI-SUBJECT IMAGE PERSONALIZATION

A widely used method for achieving multi-subject image personalization involves training individual models for each subject, followed by their integration. Tuning-based methods (Gu et al., 2023; Avrahami et al., 2023; Liu et al., 2023b; Kumari et al., 2023) with improvements on visual conflicts can produce impressive multi-subject personalized images. However, zero-shot methods eliminate the need for individual subject tuning or the merging of different combinations, significantly reducing costs and enhancing the practicality of multi-subject personalization. This makes research into zero-shot multi-subject image personalization both necessary and promising.

Most of the relevant studies (Ma et al., 2024; Gu et al., 2023; Liu et al., 2023b; Xiao et al., 2024) focus on mitigating visual conflicts in text cross-attentions of the base model. While modifying text cross-attention can be effective, it presents certain limitations. First, adjustments to text cross-attention can directly impact the control over text conditions. Second, text cross-attention does not directly dictate the areas of influence for image conditions; rather, it exerts an indirect influence on image conditions by shaping the image layout generated by the diffusion model. This indirect control may result in low performance and increased uncertainty.

### 3.3 DATA CONSTRUCTION

In the field of multi-subject personalization, creating a robust dataset architecture is a significant challenge, especially when no pre-existing dataset includes a variety of reference subjects with their validated truths. Our method starts with applying a Named Entity Recognition (NER) protocol to textual data to extract relevant entities. These entities are then used within a detection framework to define the corresponding bounding boxes. This step generates training samples that encompass a range of [*subjects, entities, spatial layouts*].

Previous studies have mostly created training examples from stand-alone images which is essentially a reconstruction task, leading the resulting models to favor replication, often resulting in 'copy-and-paste' artifacts (Chen et al., 2023b). To address this issue, our enhanced approach involves extracting subjects from a single video frame and using another frame from the same sequence as a reference for the truth. This technique effectively separates the personalized references from the target images. Due to possible variations in subjects between frames, we use a specialized subject-matching algorithm to ensure accurate matching. We provide a detailed description of this data processing pipeline in Section A.

### 3.4 GROUNDING RESAMPLER

Different from text embedding, image embedding generally contains more information and is sparser, making projection into the condition space challenging. Leveraging embeddings from all image patches primarily control the condition, but the pooled output from the image encoder tends to omit many details. Different from Flamingo (Alayrac et al., 2022) and IP-Adapter (Ye et al., 2023), we propose the integration of a grounding resampler that functions as an alternative form of image projector. Utilizing a set of learnable tokens, a resampler queries and distills pertinent information from the image features. Specifically, with an image embedding $f_i$ and a learnable query $f_q$, the resampler comprises several attention layers:

$$\text{RSAttn} = \text{Softmax}\left(\frac{\mathbf{Q}\left(f_q\right)\mathbf{K}^\top\left([f_i, f_q]\right)}{\sqrt{d}}\right)\mathbf{V}\left([f_i, f_q]\right) \qquad (3)$$

where $[f_i, f_q]$ denotes the concatenation of the image embedding $f_i$ and the learnable query $f_q$. The architecture incorporates fully connected feedforward networks (FFNs), analogous to those utilized in standard vision transformers (Dosovitskiy et al., 2021).

As detailed in 3.3, we can obtain entities of multiple referenced subjects and their target area boxes in the generated image. We present to initialize the query $f_q$ with grounding tokens $f_g$ derived from text embedding of entities and Fourier embedding of boxes. Entities are related to the semantic information of images and boxes indicate the areas where the subjects are supposed to be. It would be helpful for the resampler to extract the image features appropriately and the cross-attention layers to condition the generation finely. To prevent the model from becoming dependent on the grounding tokens during inference, we randomly replace them with the original learnable queries in the training. For the input of n subjects, the projection processes of different subject images do not affect each other. The resulting n queries will be concatenated and input into the subsequent model as $\mathbf{c_i}$ with $N = n * n_t$ tokens, where $n_t$ is the token quantity per subject.

### 3.5 MULTI-SUBJECT CROSS-ATTENTION

In scenarios involving the generation of multiple subjects, challenges frequently arise that are not exclusive to personalization tasks. These include discordances between subjects and their backgrounds and amongst the subjects themselves. A viable solution to mitigate such conflicts leverages attention masks, contingent upon the availability of layout priors. The incorporation of attention masks within cross-attention mechanisms facilitates the exclusion of padding tokens from the condition, thus minimizing their impact.

To confine the context of each subject to a designated spatial domain, we propose an enhancement to the conventional attention mask, denoted as $\mathbf{M}$. This adjustment involves the bilateral neglection of

tokens within both the query and key matrices, applied specifically for the $j$th subject as follows:

$$\mathbf{M}_j(x, y) = \begin{cases} 0 & \text{if } [x, y] \in B_j \\ -\infty & \text{if } [x, y] \notin B_j \end{cases} \tag{4}$$

Here, $B_j$ signifies the coordinate set of bounding boxes pertaining to the $j$th subject. By this means, the conditional image latent $\hat{\mathbf{z}}_{img}$ is derived through:

$$\hat{\mathbf{z}}_{img} = \text{Softmax}\left(\frac{\mathbf{Q}\mathbf{K}_i^\top}{\sqrt{d}} + \mathbf{M}\right)\mathbf{V}_i \tag{5}$$

Herein, $\mathbf{M}$ represents the amalgamation of all subject-specific masks, $\text{Concat}(\mathbf{M}_0, \ldots, \mathbf{M}_n)$. In this way, the model ensures each subject to be represented in a certain area, thus resolving the issues of subject neglect and conflict in Figure 2.

However, an inherent limitation arises when a query patch token is ubiquitously masked across all referenced subjects or remains unmasked (in instances of overlapping bounding boxes), thereby diminishing the intended efficacy of multi-subject cross-attention. To counteract this, we introduce dummy tokens initialized at random preceding the image tokens to symbolize the background. This strategy is instrumental in ensuring that text conditions predominantly govern areas devoid of any guided layout, thereby solving the subject overcontrol issue in Figure 2. Following the acquisition of $\mathbf{A}$, we apply $\mathbf{M}_{bg}$ to seamlessly mask these tokens within $\hat{\mathbf{z}}_{img}$, as illustrated:

$$\mathbf{z}_{img} = (1 - \mathbf{M}_{bg}) \cdot \hat{\mathbf{z}}_{img} \tag{6}$$

where $\mathbf{M}_{bg}$ is articulated as a binary mask, with elements within the subject bounding boxes designated as zero. In contrast to the methods discussed in Section 3.2, our proposed multi-subject cross-attention directly manages the image conditions by employing masked image cross-attention in the targeted areas. While addressing multi-object conflicts, our method ensures that text conditions remain unaffected, as evidenced by the significantly higher text adherence capability shown in Table 2. Notably, certain studies have sought to resolve these conflicts through the application of objectives on attention maps within the cross-attention mechanism. A series of rigorous experiments have been conducted to substantiate our design, with the details elucidated in Section 4.4.

## 4 EXPERIMENTS

### 4.1 EXPERIMENT SETUP

**Datasets.** For training, we utilize an in-house video dataset that contains 3.6M video clips. For evaluation, we measure the single-subject and multi-subject performance on DreamBench (Ruiz et al., 2023) and MS-Bench, respectively. DreamBench includes 30 subjects and 25 prompts and we preset all input boxes to [0.25, 0.25, 0.75, 0.75]. To thoroughly assess the performance of multi-subject personalization, we have established a new evaluation standard, MS-Bench, which includes 40 subjects and 1148 combinations, each combination having up to 6 prompts, totaling 4488 distinct test samples. Details of datasets are provided in Section A and Section B.

**Evaluation metrics.** Following previous works, we measure the performance through image and text fidelity. To assess image fidelity, we employ cosine similarity measures between generated images and subject images within CLIP (Radford et al., 2021) and DINO (Caron et al., 2021) spaces, referred to as CLIP-I and DINO, respectively. For text fidelity, we utilize cosine similarity between generated images and text prompts in CLIP space, denoted as CLIP-T. In multi-subject personalization, using the average fidelity to reflect image fidelity is insufficient, as it fails to reveal cases of subject neglect. We further employ the product of multi-subject DINO, denoted as M-DINO, to indicate whether each subject has been recreated in the results.

**Baselines.** For single-subject personalization, we compare our model with methods mentioned in Table 1. Emu2 (Sun et al., 2023), Kosmos-G (Pan et al., 2023), and $\lambda$-ECLIPSE (Patel et al., 2024) are all MLLM-based methods, while $\lambda$-ECLIPSE is reported to have better performance. Therefore, we select SSR-Encoder (Zhang et al., 2024) and $\lambda$-ECLIPSE (Patel et al., 2024) as baselines for multi-subject personalization. Considering the fairness, the qualitative results of MS-Diffusion are generated without any fine-tuning on benchmarks. The implementation details of MS-Diffusion and these methods are contained in Section C.

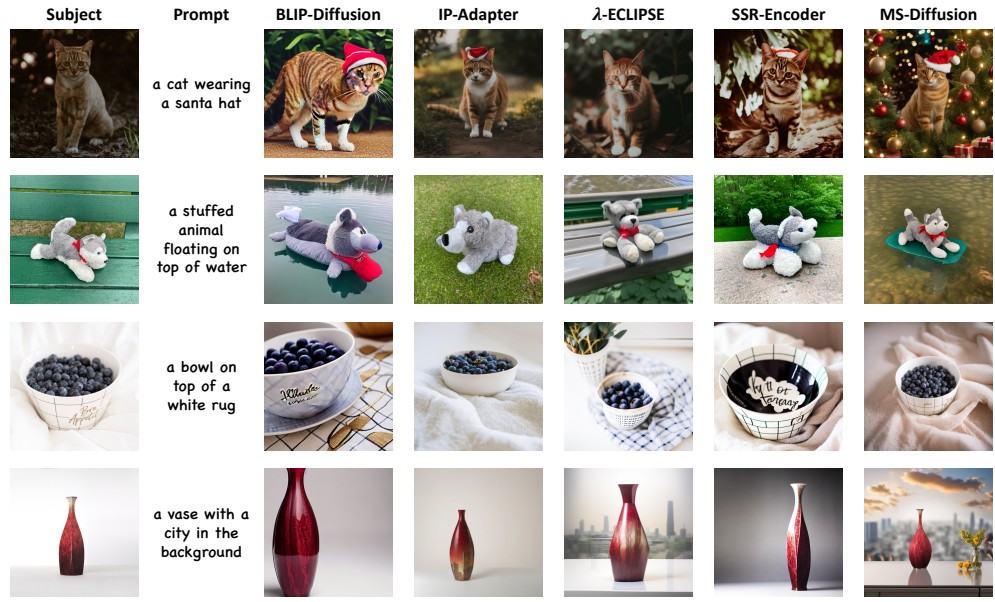

Figure 4: Qualitative results of MS-Diffusion and baselines in single-subject personalization.

Table 2: **Quantitative comparison on MS-Diffusion and baselines. Bold** and underline represent the highest and second-highest metrics, respectively. For single-subject, the results for IP-Adapter, Emu2, and Kosmos-G were obtained from λ-ECLIPSE (Patel et al., 2024), while the rest are reported in the corresponding papers. * denotes the model is fine-tuned on DreamBench.

| Method | Single-subject | | | Multi-subject | | | |
|---|---|---|---|---|---|---|---|
| | CLIP-I | DINO | CLIP-T | CLIP-I | DINO | M-DINO | CLIP-T |
| Textual Inversion | 0.780 | 0.569 | 0.255 | - | - | - | - |
| DreamBooth | 0.803 | 0.668 | 0.305 | - | - | - | - |
| BLIP-Diffusion* | **0.805** | 0.670 | 0.302 | - | - | - | - |
| λ-ECLIPSE* | 0.796 | 0.682 | 0.304 | - | - | - | - |
| **MS-Diffusion*** | **0.805** | **0.702** | **0.313** | - | - | - | - |
| ELITE | 0.771 | 0.621 | 0.293 | - | - | - | - |
| BLIP-Diffusion | 0.779 | 0.594 | 0.300 | - | - | - | - |
| IP-Adapter | 0.810 | 0.613 | 0.292 | - | - | - | - |
| Emu2 | 0.765 | 0.563 | 0.273 | - | - | - | - |
| Kosmos-G | **0.822** | 0.618 | 0.250 | - | - | - | - |
| λ-ECLIPSE | 0.783 | 0.613 | 0.307 | 0.724 | 0.419 | 0.094 | 0.316 |
| SSR-Encoder | 0.821 | 0.612 | 0.308 | **0.725** | **0.425** | 0.107 | 0.303 |
| **MS-Diffusion** | 0.792 | **0.671** | **0.321** | 0.698 | **0.425** | **0.108** | **0.341** |

## 4.2 SINGLE-SUBJECT COMPARISON

In the single-subject comparison, a detailed examination is carried out utilizing both qualitative and quantitative comparisons to gauge the performance of different methodologies. On the qualitative front, as shown in Figure 4, MS-Diffusion shows an exceptional ability to generate single-subject images with high fidelity and detail retention. In quantitative results provided in Table 2, MS-Diffusion also achieves competitive scores, with obviously the highest DINO and CLIP-T scores at 0.671 and 0.321 respectively on zero-shot scenarios, and leading CLIP-I score of 0.792. For tuning methods, MS-Diffusion outperforms baselines in all metrics. As discussed in DreamBooth (Ruiz et al., 2023), DINO more accurately captures the similarity in details between the results and the

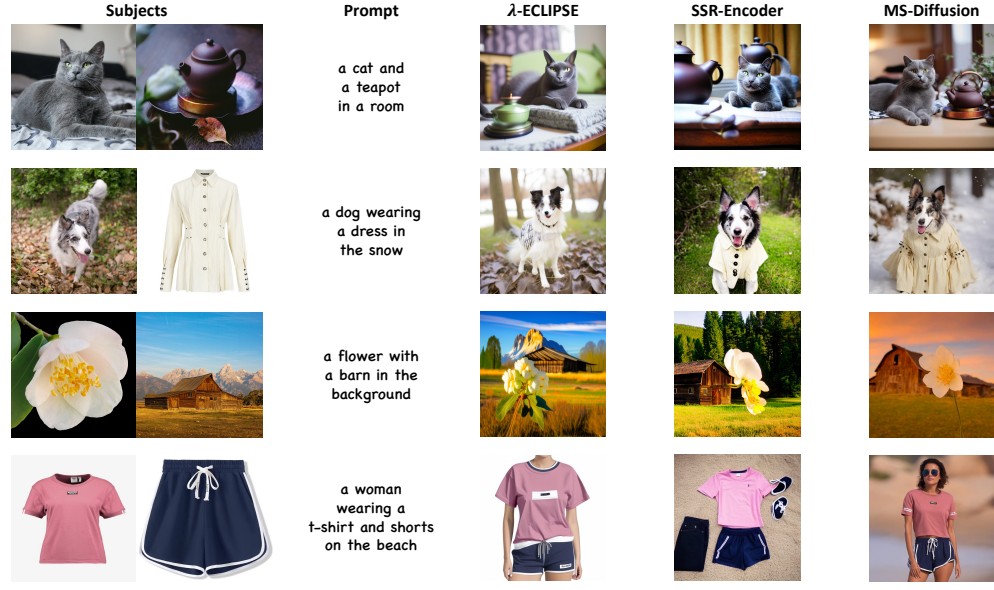

Figure 5: Qualitative results of MS-Diffusion and baselines in multi-subject personalization.

Table 3: **Ablation study of MS-Diffusion.** RS, GRS, MCA, TAL, IAL, and LG represent resampler, grounding resampler, multi-subject cross-attention, text attention loss, image attention loss, and layout guidance, respectively.

| Method | Single-subject | | | Multi-subject | | | |
|---|---|---|---|---|---|---|---|
| | CLIP-I | DINO | CLIP-T | CLIP-I | DINO | M-DINO | CLIP-T |
| **MS-Diffusion** | 0.792 | **0.671** | **0.321** | **0.698** | **0.425** | **0.108** | **0.341** |
| w/o RS | 0.775 | 0.583 | 0.320 | 0.680 | 0.372 | 0.082 | 0.336 |
| w/o GRS | 0.777 | 0.646 | 0.320 | 0.681 | 0.389 | 0.090 | 0.331 |
| w/o MCA | 0.798 | 0.662 | 0.312 | 0.693 | 0.422 | 0.100 | 0.309 |
| w/o LG w/ IAL | 0.761 | 0.577 | 0.284 | 0.675 | 0.377 | 0.080 | 0.305 |
| w/o LG w/ IAL&TAL | **0.809** | 0.660 | 0.293 | 0.687 | 0.413 | 0.093 | 0.316 |

labels, whereas CLIP-I may exhibit high scores in situations of background overfitting, resulting in a clear advantage for DINO, but a slight disadvantage for CLIP-I of MS-Diffusion.

## 4.3 MULTI-SUBJECT COMPARISON

From a qualitative perspective in Figure 5, MS-Diffusion manages to maintain natural interactions among subjects in generated images while ensuring each subject retains its distinctiveness and recognizability. Quantitatively, results in Table 2 demonstrate the strength of MS-Diffusion in DINO, M-DINO, and CLIP-T. Unlike in single-subject personalization, there is a larger gap in text fidelity between MS-Diffusion and the baselines in multi-subject personalization, demonstrating that MS-Diffusion not only effectively generates the multiple subjects outlined in the text but also excellently preserves the text control capabilities inherent to SD. Additionally, the image fidelity of MS-Diffusion is comparable, highlighting its superior ability to retain details, particularly significant as low text fidelity is commonly associated with overfitting.

## 4.4 ABLATION STUDY

**Module ablation.** We conduct an ablation experiment on the proposed two modules, grounding resampler (GRS) and multi-subject cross-attention (MCA), to validate their effects. For GRS, we replace it with a linear projection layer and a normal resampler (Alayrac et al., 2022; Ye et al., 2023).

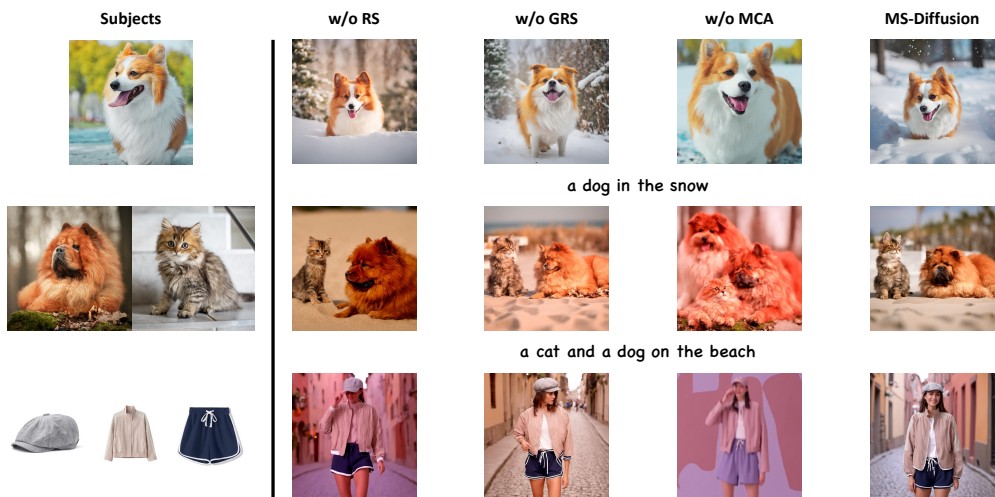

Figure 6: **Visualization results of module ablation.** Models without RS or GRS have an obvious decrease in the detail-preserving capability. For the multi-subject generation, the model without MCA cannot handle the subject conflicts.

Results in Table 3 indicate that the resampler-like image projector significantly enhances the details, as evidenced by DINO being obviouly higher than the linear projector. Moreover, The substantial improvement in multi-object image fidelity by GRS reflects the critical role of the information carried by grounding tokens in multi-object generation. As a key module for resolving conflicts, removing MCA results in a noticeable degradation of text fidelity, especially in multi-subject generation. The combined use of both modules ensures that MS-Diffusion maintains high image and text fidelity simultaneously. We provide visualization results regarding the module ablation in Figure 6. As clearly reflected by the qualitative examples, GRS enhances the details and MCA handles the conflicts.

**Layout guidance.** As mentioned in Section 3.5, we have explored the indispensable role of explicit layout guidance (LG), including grounding tokens and MCA. A straightforward approach to implicitly utilizing layout involves incorporating an attention loss during training. Besides the image attention loss (IAL), we also introduce text attention loss (TAL) to training by setting the original cross-attention layers trainable. The detailed loss definition is provided in Section G. As illustrated in Table 3, an objective to guide the image cross-attention helps the personalization hardly at all. TAL has somewhat resolved the conflict issues, but its performance is inferior to MS-Diffusion while introducing additional training parameters. We consider the inclusion of LG necessary and rational, not merely for the performance enhancements it offers, but also because it effectively resolves the various multi-object generation issues highlighted in Figure 2.

## 5    CONCLUSION

This study makes a significant contribution to the field of P-T2I diffusion models with the development of MS-Diffusion. This zero-shot framework excels at capturing intricate subject details and smoothly blending multiple subjects into a single coherent image. Equipped with the innovative Grounding Resampler and Multi-subject Cross-attention mechanisms, our model effectively overcomes common multi-subject personalization issues, such as subject neglect and conflict. Extensive ablation studies underscore MS-Diffusion's enhanced performance in image synthesis fidelity compared to existing models. It stands as a groundbreaking approach for P-T2I applications that are free from the need for fine-tuning and require layout guidance.

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

## A    TRAINING DATASET CONSTRUCTION PIPELINE

Figure 7 illustrates the construction pipeline of the training dataset. Firstly, we randomly select two frames from a video clip, one as the reference and the other as the ground truth. Both frames are captioned by BLIP-2 (Li et al., 2023b). Secondly, we utilize a NER model[1] to extract entities from the caption. Entities and images are then input into Grounding DINO (Liu et al., 2023a) to obtain the boxes, which are parts of the final input of the model. Taking the boxes as prompts of SAM (Kirillov et al., 2023), we can further obtain segmentation masks to extract subjects from the reference image. Since the entities in different frames can be different, we design a subject matcher, which finds the correspondence between the frames by conducting Hungarian Algorithm on the entity image embeddings. Frames in a video typically contain the same entities but exhibit clear differences in details such as angles and poses. This makes them highly suitable as training data for personalized image generation models, which can help mitigate the model's tendency to copy-and-paste.

Our dataset comprises 2.8M general scenario videos and 0.8M product demonstration videos, where the former covers more scenarios and the latter has more clear subjects. 2-5 frames for each are adopted in the training. In practice, there may only be 1-2 subjects successfully matched. To ensure that the training data contains a sufficient number of reference subjects, for the targets where matching fails, we directly use the corresponding parts of the ground truth as references. Subjects that are too small, too large, or have imbalanced proportions are filtered out. Each training sample can have up to 4 subjects, and we pad in the ones with fewer than 4.

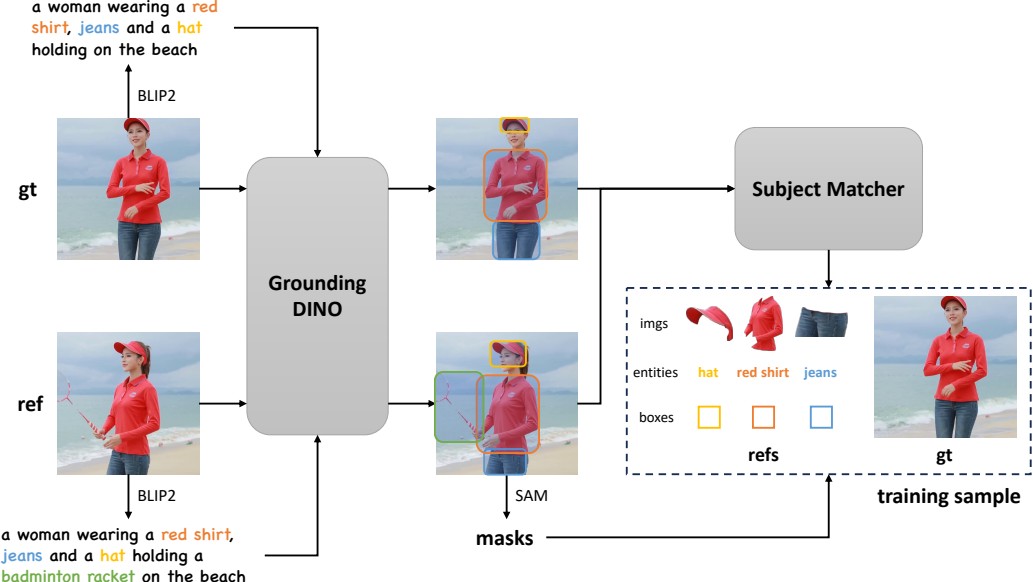

Figure 7: **Data construction pipeline of our work.** For the input of two frames, we can get subject images, entities, and boxes. Note that the entities and boxes are from the ground truth frame since they indicate the information in the generated result.

## B    DETAILS OF MS-BENCH

To construct MS-Bench, we collect subjects from previous studies (Ruiz et al., 2023; Gal et al., 2023; Kumari et al., 2023), the Internet[2], and an internal dataset that does not overlap with the training set. MS-Bench contains four data types and 13 combination types with two or three subjects. We provide the details in Table 4. Each combination type other than those related to the scene has 6 prompt variations. There are 1148 combinations and 4488 evaluation samples, where entities and

---

[1]https://spacy.io/
[2]https://unsplash.com/

Table 4: **Explanation of MS-Bench.** Each combination type has preset prompts and boxes. **[S]** represents prompt variations about the scene, including "in a room", "in the jungle", "in the snow", "on the beach", "on the grass", and "on a cobblestone street".

| Type | Prompt | Boxes |
|---|---|---|
| living+living
living+object
object+object | a {0} and a {1} [S] | [0.00, 0.25, 0.50, 0.75]
[0.50, 0.25, 1.00, 0.75] |
| living+upwearing | a {0} wearing a {1} [S] | [0.25, 0.25, 0.75, 0.75]
[0.25, 0.00, 0.75, 0.25] |
| living+midwearing
living+wholewearing | a {0} wearing a {1} [S] | [0.25, 0.25, 0.75, 0.75]
[0.25, 0.25, 0.75, 0.75] |
| midwearing+downwearing | a woman wearing a {0}
and a {1} [S] | [0.25, 0.25, 0.75, 0.60]
[0.25, 0.60, 0.75, 1.00] |
| living+scene
object+scene | a {0} with a {1}
in the background | [0.25, 0.25, 0.75, 0.75]
[0.00, 0.00, 1.00, 1.00] |
| living+living+living
object+object+object | a {0}, a {1}, and a {2} [S] | [0.00, 0.25, 0.35, 0.75]
[0.35, 0.25, 0.65, 0.75]
[0.65, 0.25, 1.00, 0.75] |
| living+object+scene | a {0} and a {1} with a {2}
in the background | [0.00, 0.25, 0.50, 0.75]
[0.50, 0.25, 1.00, 0.75]
[0.00, 0.00, 1.00, 1.00] |
| upwearing+midwearing+
downwearing | a woman wearing a {0}, a {1},
and a {2} [S] | [0.25, 0.00, 0.75, 0.25]
[0.25, 0.25, 0.75, 0.60]
[0.25, 0.60, 0.75, 1.00] |

boxes are subject categories and preset layouts. Compared to other multi-subject benchmarks, our MS-Bench ensures that the model performance can be reflected comprehensively in abundant cases.

## C EXPERIMENT SETTINGS

**Training and Inference.** The pre-trained model employed in MS-Diffusion is Stable Diffusion XL (SDXL) (Podell et al., 2023). Implemented by Pytorch 2.0.1 and Diffusers 0.23.1, our model is trained on 16 A100 GPUs for 120k steps with a batch size of 8 and a learning rate of 1e-4. Following the training of IP-adapter (Ye et al., 2023), we set $\gamma = 1.0$ in cross-attention layers and dropped the text and image condition using the same probability. To ensure the model is not dependent on the grounding tokens (Section 3.4), we also randomly drop them with a probability of 0.1. We generate five images for each sample during the inference, with unconditional guidance scale and $\gamma$ set to 7.5 and 0.6, respectively, to get better results.

**Comparative methods.** Here we provide the details of the baselines compared in qualitative and quantitative experiments:

- **BLIP-Diffusion** (Li et al., 2023a) utilizes BLIP-2 (Li et al., 2023b) to unite the text and image embeddings. We implement it in the qualitative comparison using Diffusers.

- **IP-Adapter** (Ye et al., 2023) also uses image prompt as the condition. We run qualitative samples on their official code with the scale set to 0.5 recommended by the paper. Considering fairness, we use the result of SDXL in Table 2.

- **SSR-Encoder** (Zhang et al., 2024) design a query network to extract the specified subject of a single image, which enables it to finish multi-subject generation. We leverage it as one of the baselines in multi-subject personalization. For performance comparison, we

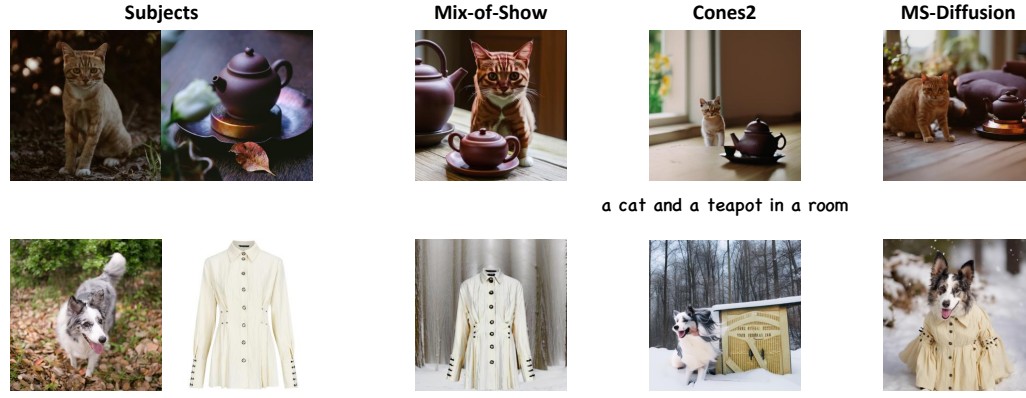

Figure 8: **Qualitative comparison of Mix-of-Show, Cones2, and MS-Diffusion.** MS-Diffusion outperforms Mix-of-Show and Cones2 in details preserving in the first row. Mix-of-Show and Cones2 struggle in handling the interactions between subjects.

employed the official code provided, alongside the default hyperparameters specified in the code repository.

- $\lambda$-**ECLIPSE** (Patel et al., 2024) trains an independent multi-modal encoder and employs Kandinsky as the generative backbone. Since it outperforms other MLLM-based approaches, we choose it to be the representative. To facilitate a comparative analysis of performance, the study utilized the officially provided code, in conjunction with the default hyperparameters delineated within the corresponding code repository.

## D    MORE RESULTS OF SINGLE-SUBJECT PERSONALIZATION

Additional qualitative results on DreamBench are provided in Figure 12. MS-Diffusion shows excellent text fidelity in all subjects while keeping subject details, especially the living ones (dogs). It can be noticed that some elements in the background (the third line and the fourth line) also occur in the results (the grass and the teapot holder) since the entire images are referenced during the generation. Their scope of action depends on the input bounding box. In practical applications, using masked images as a condition is recommended.

One of the limitations in detail preservation is the insufficient capability of the image encoder. We provide some uncommon examples in Figure 13. MS-Diffusion utilizes the CLIP image encoder, which results in the loss of some details in uncommon and complex cases. However, it still significantly outperforms state-of-the-art methods benefiting from the proposed grounding resampler.

## E    MORE RESULTS OF MULTI-SUBJECT PERSONALIZATION

We provide additional multi-subject personalized images based on MS-Bench in Figure 14. The results encompass various combination types, fully demonstrating the generalizability and robustness of MS-Diffusion. When the scene changes freely according to the text, the details of the subject are preserved without being affected. In addition to common parallel combinations, MS-Diffusion also performs well in personalized generation for combinations with certain overlapping areas, such as "living+midwearing" and "object+scene".

## F    COMPARISON WITH TUNING-BASED METHODS

While zero-shot methods like MS-Diffusion can decrease the tuning cost, they may also suffer from performance degradation compared to tuning-based approaches due to the limitations of pre-training scale. However, MS-Diffusion indicates comparable performance in single-subject quantitative results

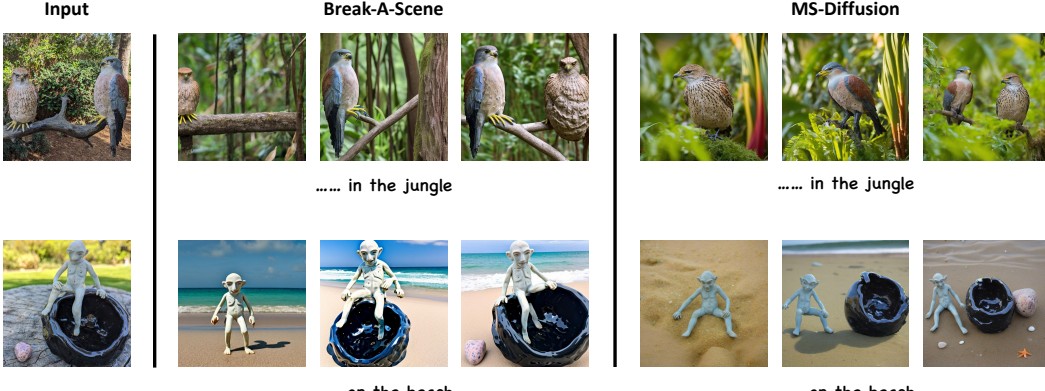

Figure 9: **Qualitative comparison of Break-A-Scene and MS-Diffusion.** MS-Diffusion gets comparable results by extracting subjects from a single image. Break-A-Scene tends to overfit the input image (the bird pose in the first row and the white creature sitting on the black bowl in the second row).

in Table 2. Since the proposed MS-Bench can be too large for tuning-based methods, we provide qualitative comparison results with Mix-of-Show (Gu et al., 2023), Cones2 (Liu et al., 2023b), and Break-A-Scene (Avrahami et al., 2023) in Figure 8 and Figure 9. The results show that MS-Diffusion achieves comparable results to tuning-based methods. Mix-of-Show and Cones2 face certain issues when handling multiple subjects with complex interactions (e.g., the example of a dog wearing a dress). Break-A-Scene tends to overfit the original image's interactions (e.g., the pose of birds and the white creature sitting on the black bowl). While avoiding these issues, MS-Diffusion requires no test-time tuning and only one subject image during inference, unlike the tuning-based methods that need additional time and multiple images for tuning.

## G    LAYOUT GUIDANCE

Cross-attention maps can intuitively reflect the condition-image attribution relation (Tang et al., 2023; Hertz et al., 2023). Recent works (Wang et al., 2023; Wei et al., 2024) have studied utilizing an objective on the cross-attention maps in multi-subject generation. The objective exists only in training, considered implicit and insufficient to handle multi-subject conflicts. We have provided the performance comparison between our explicit layout guidance and attention loss in Section 4.4. For a single cross-attention layer, the attention loss of the $j$th subject is calculated by:

$$\mathcal{L}_{am}^j = \left(1 - \frac{\sum_{[x,y] \in B_j} \mathbf{A}_{[x,y],j}}{\sum_{[x,y]} \mathbf{A}_{[x,y],j}}\right)^2 \tag{7}$$

where $[x,y]$ corresponds to a latent token in $\mathbf{Q}$ and $B_j$ is the bounding box of the $j$th subject. This objective aims to promote the activation of attention maps within specific boxes. We average $\mathcal{L}_{am}^j$ across layers and subjects and set its weight to 0.01 in the final loss. To validate the text attention loss, we also optimize the text cross-attention layers in training, increasing approximately 70% in learnable parameters.

Although our model provides explicit layout guidance, it still significantly differs from layout-based diffusion. Firstly, the information of boxes in the grounding resampler is prior, and its conditional effect is relatively weak. We have also reduced the model's reliance on this input by randomly dropping grounding tokens. Secondly, the multi-subject cross-attention only exists in image cross-attention, inherently controlling the action of image conditions in specified areas, but cannot determine the whole generation of the diffusion model. Our goal is not to develop a method that fully supports layout control but to utilize layout information to guide the model in resolving conflicts in multi-subject generation.

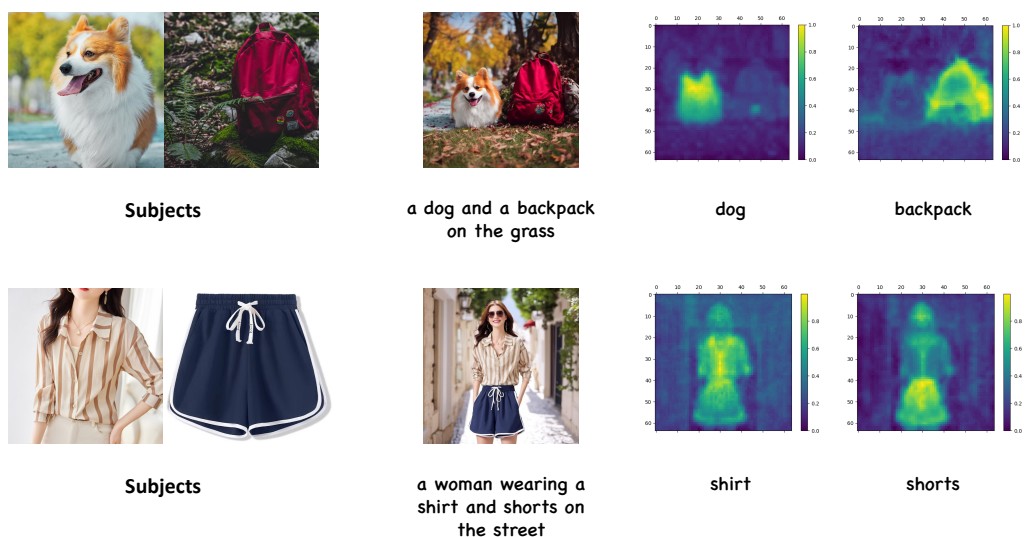

Figure 10: **Text-image attribution analysis of MS-Diffusion.** We average the attention maps corresponding to the subjects and translate them to normalized heat maps.

To further explore the layout control capability of MS-Diffusion, we provide qualitative results in Figure 15. It can be demonstrated that MS-Diffusion can generate images that adhere to layout conditions, even in the case of two instances of the same category. However, the generated positions are not entirely accurate, especially in *"a cat and a cat on the grass"*, illustrating that the layout condition is relatively weak compared to text and image prompts in the personalization task.

## H    INTEGRATION WITH CONTROLNET

In the realm of text-to-image diffusion models, a notable application is the enhancement of structural control within image generation. Our MS-Diffusoin maintains the original network architecture unchanged, thus ensuring full compatibility with existing controllable tools. Consequently, this allows for the generation of images that are not only prompted by images but also governed by additional conditions. By integrating our MS-Diffusion with established controllable mechanisms such as ControlNet, we demonstrate the capacity to produce images under varied structural directives. Figure 16 displays an array of images synthesized with image prompts coupled with distinct structural controls(depth, canny edge, openpose). This seamless cooperation between our method and these tools facilitates the creation of highly controllable images without necessitating fine-tuning.

## I    MULTI-SUBJECT INTERACTION

Unlike image editing (Kawar et al., 2023; Brooks et al., 2023), personalized image generation features a high degree of freedom, enabling effective handling of interactions among different elements. Benefiting from its architecture design that does not impact the base model, MS-Diffusion successfully inherits the multi-subject interaction capabilities of the base model. As illustrated in Figure 17, MS-Diffusion can flexibly manage interactions between reference subjects, even when there is overlap among these objects, thereby demonstrating its potential in practical applications.

## J    HUMAN AND ANIME PERSONALIZATION

Human and anime subjects are popular in the use of personalized model. We provide results of MS-Diffusion on human and anime subjects in Figure 18. Some research (Wang et al., 2024a; Li et al., 2024) has explored human personalization in text-to-image diffusion models. By utilizing a

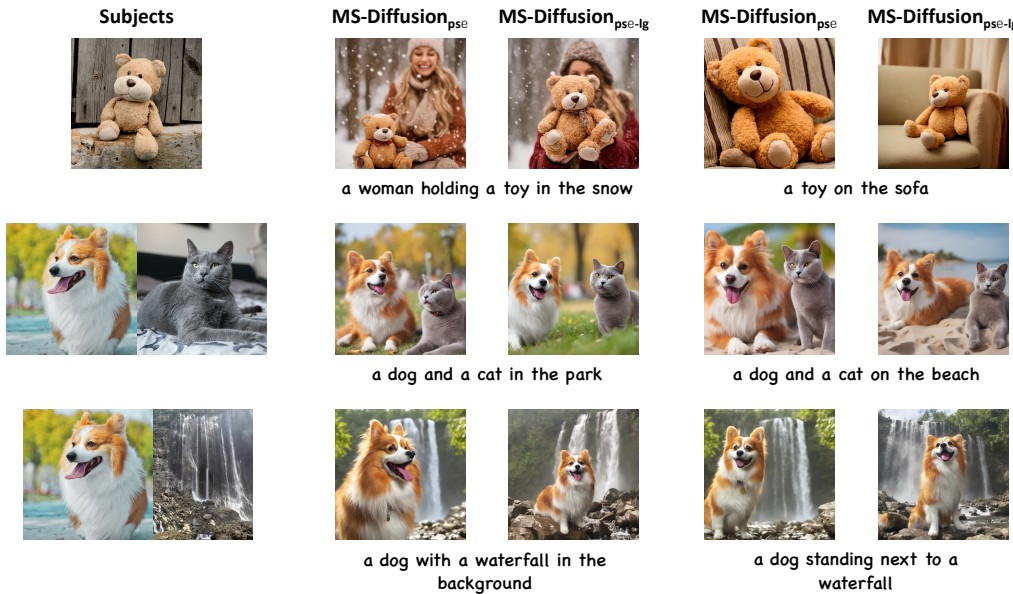

Figure 11: **Qualitative examples when applying pseudo layout guidance during the inference.** In this figure, *pse-lg* and *pse* respectively indicate whether layout prior is used.

face encoder (Deng et al., 2019) and training on the face dataset, MS-Diffusion can also be extended to a personalized model for these subjects.

## K   TEXT-IMAGE ATTRIBUTION ANALYSIS

In MS-Diffusion, our focus is primarily on resolving conflicts between subjects without altering the control mechanism of the text. While some approaches (Wang et al., 2023; 2024b; Zhou et al., 2024; Kim et al., 2023) in non-personalized text-to-image tasks address multi-object generation conflicts through text cross-attention, this inevitably requires tuning the diffusion model's parameters, thereby affecting the plug-and-play nature, which is not preferred by us. As demonstrated in the text-image attribution analysis presented in Figure 10, the control of multiple objects by text in our model is also quite evident. This may be related to the explicit layout guidance for subjects, since the images and text condition jointly in the generation process. We also attempted to control text cross-attention using the same mechanism as in Section 3.5, but no differences were observed in the results.

## L   SUBJECT INTERPOLATION

The blending of two distinct subject representations to yield composite subjects with hybrid characteristics is feasible through the navigation of the embedding space linking the subjects. As depicted in Figure 19, linear interpolations are conducted among dog and hat representations, subsequently rendering the interpolated subject in an unaccustomed context. The visualization reveals a natural gradation of subject appearance along the interpolated trajectory that harmonizes with the surrounding environment. This technique proves beneficial when applied in subject fusion and style transfer.

## M   LIMITATIONS

There are certain limitations in MS-Diffusion. The box-based indication of positions lacks precision, making it challenging to work effectively when the interaction between subjects is stronger. Moreover, the model requires explicit layout input during inference, and generating complex scenes becomes difficult. Though MS-Diffusion beats SOTA personalized diffusion methods in both single-subject and multi-subject generation, it still suffers from the influence of background in subject images.

We explore a solution for explicit layout needs during inference. MS-Diffusion supports using the text cross-attention maps as the pseudo layout guidance. Specifically, as indicated in Figure 10, since the text cross-attention maps can reflect the area of each text token, we can replace the layout prior with them during the inference. In practice, we set a threshold to extract masks from text cross-attention maps and apply them after $T$ denoising steps. Before $T$, we experiment with completely disabling layout guidance or using a rough box as a layout prior. The results are presented in Figure 11. Although disabling layout guidance experiences a decline in subject consistency, it still demonstrates that explicit layout guidance during inference can be optimized. One direction for exploration is to enable the model to learn the layout during training.

## N  SOCIETAL IMPACT

As an image personalization method, MS-Diffusion aims to customize images based on user-provided subjects without fine-tuning. Additionally, the multi-subject reference capability of MS-Diffusion allows users to freely combine and re-create different concepts. However, MS-Diffusion can also be used to generate deceptive images, especially those involving subject combinations that would not exist in reality, an issue that remains to be addressed in the future.

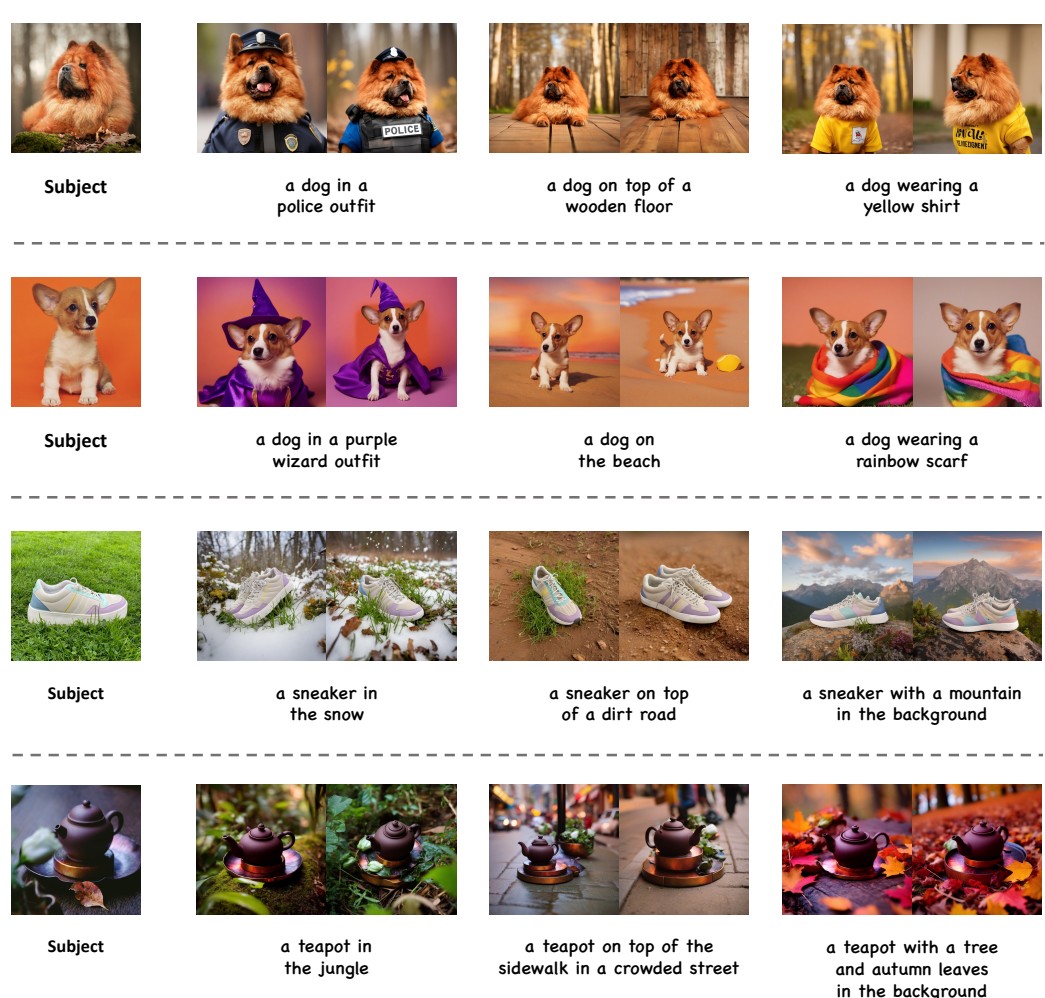

Figure 12: Additional qualitative results of MS-Diffusion in single-subject personalization.

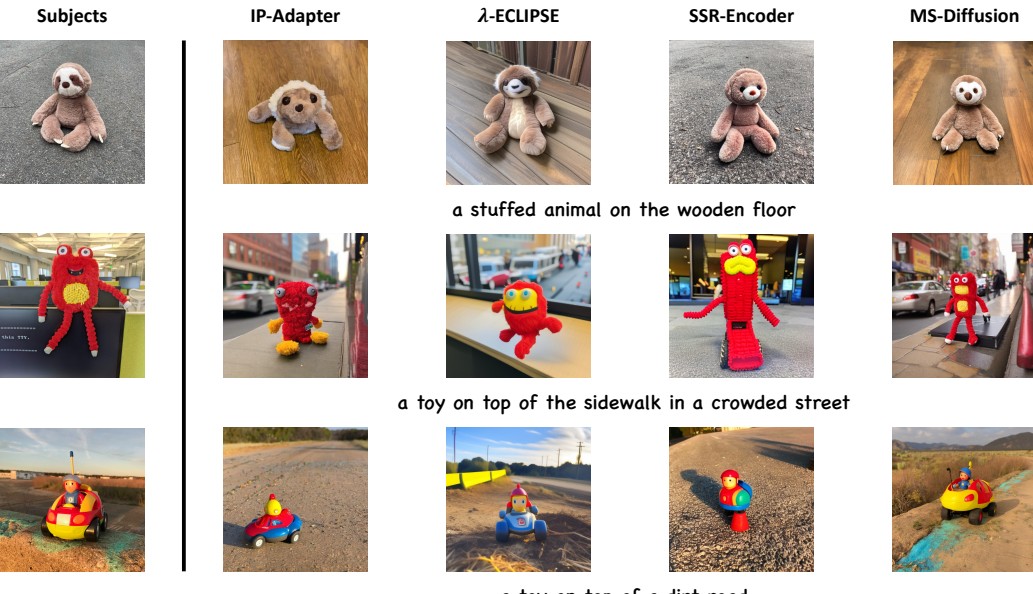

Figure 13: **Qualitative comparison of MS-Diffusion and zero-shot personalized SOTAs on uncommon subjects.** Though losing some details, MS-Diffusion outperforms other SOTAs obviously.

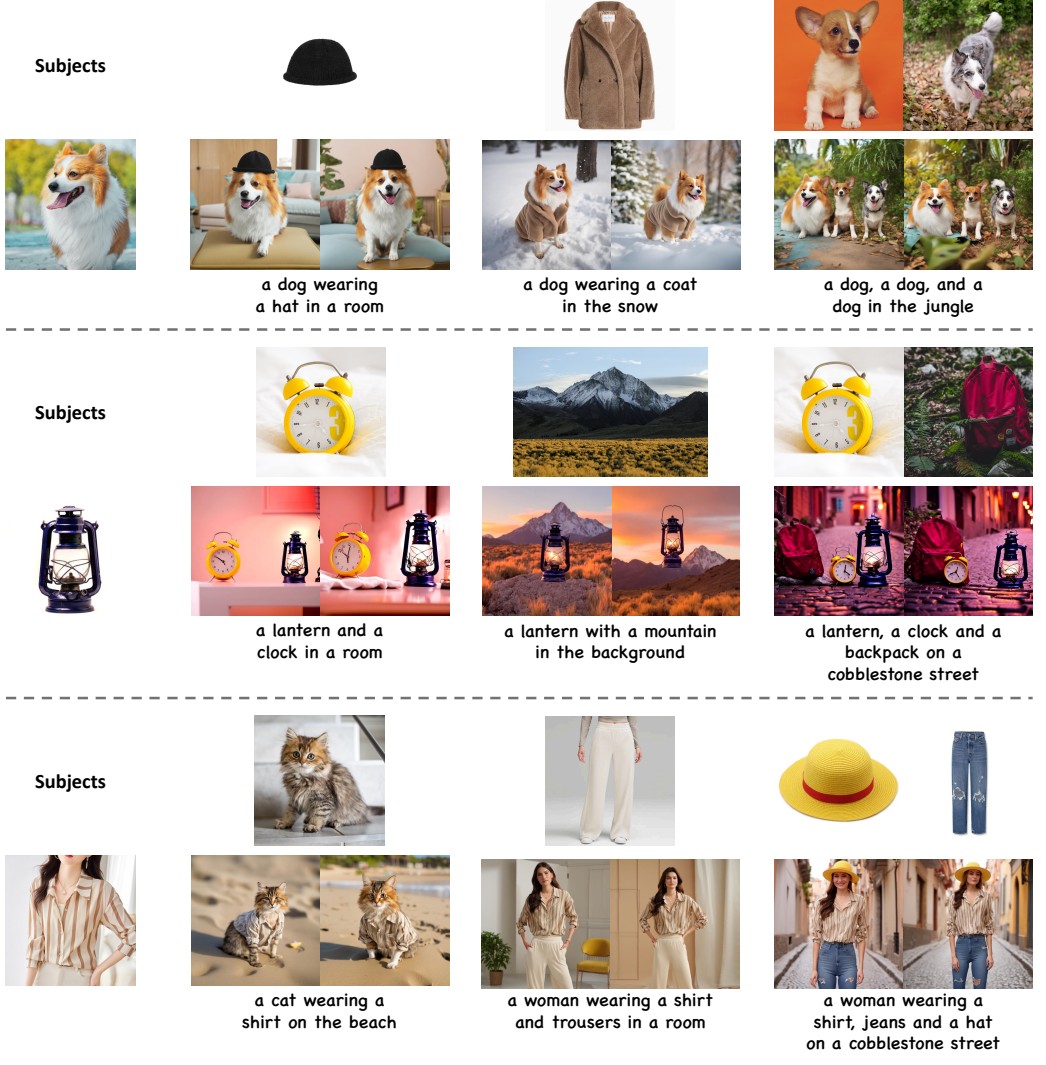

Figure 14: Additional qualitative results of MS-Diffusion in multi-subject personalization.

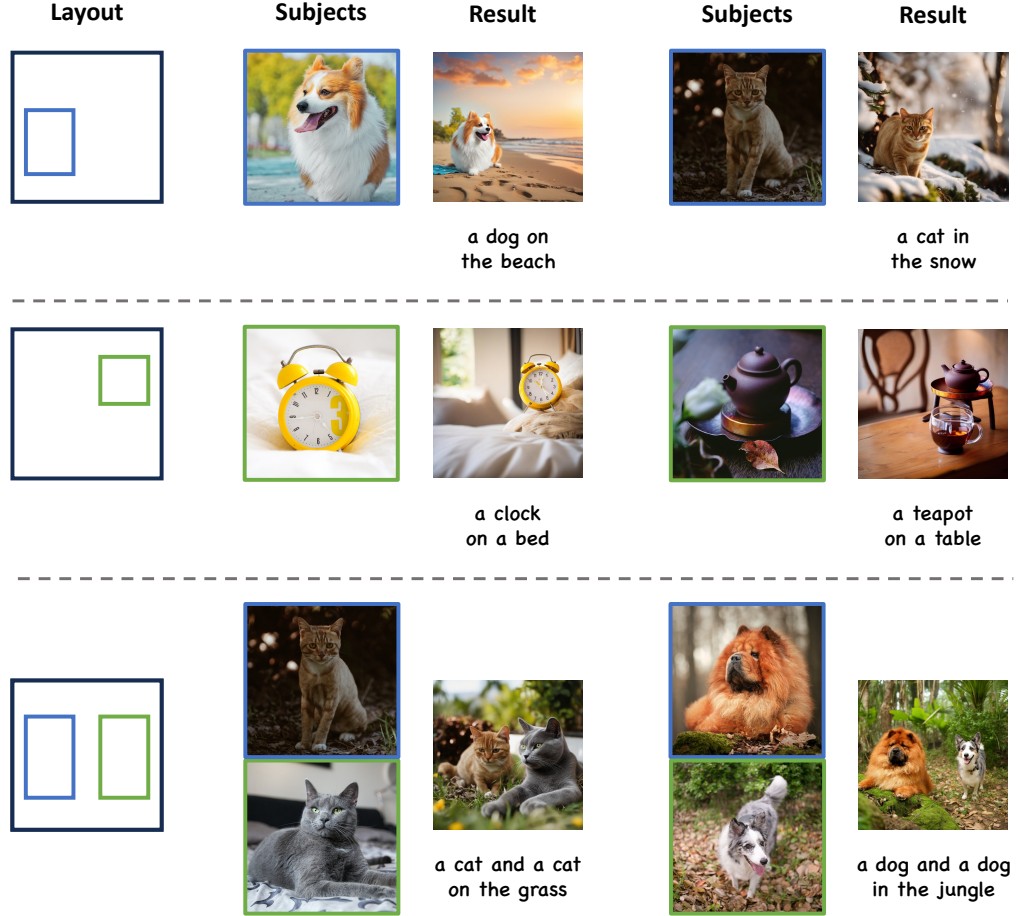

Figure 15: **Qualitative examples of MS-Diffusion about the layout control ability.** Bounding boxes of different colors correspond to subjects with different color borders.

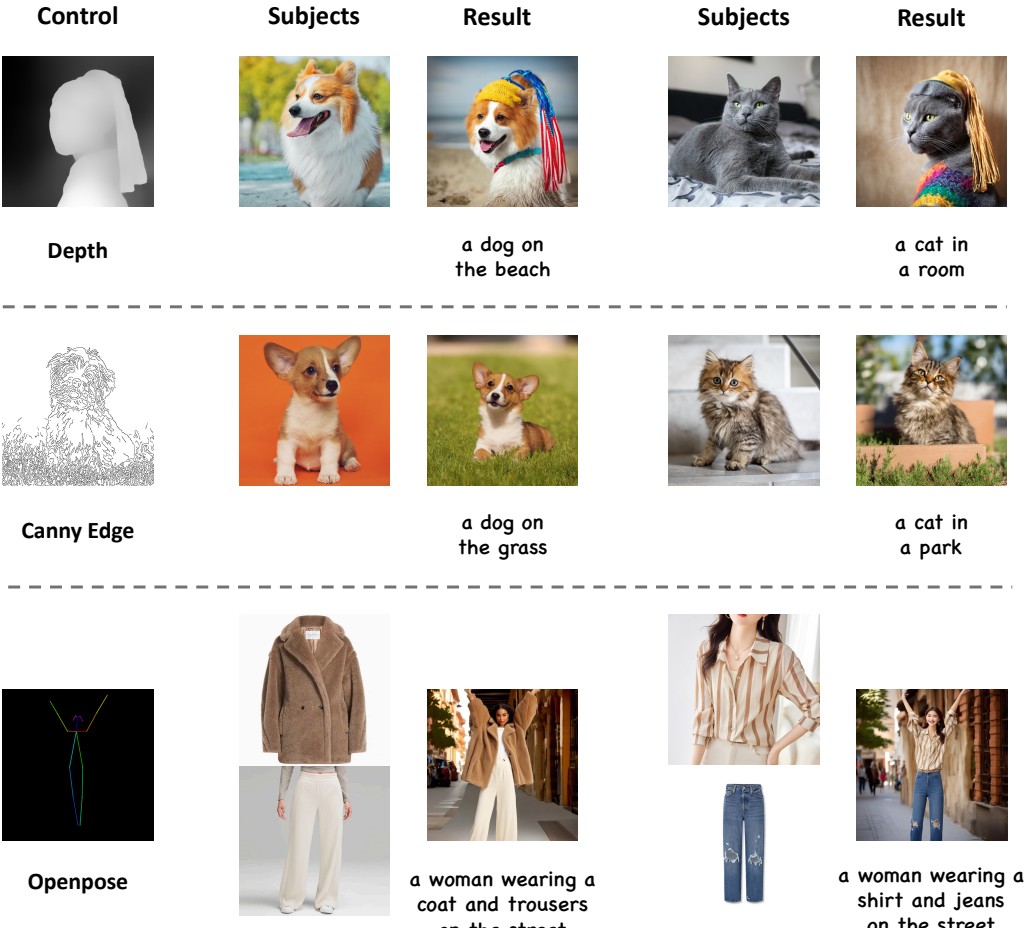

Figure 16: **Generative results when integrating different control conditions.** The integrated ControlNets are composed of depth, canny edge, and openpose.

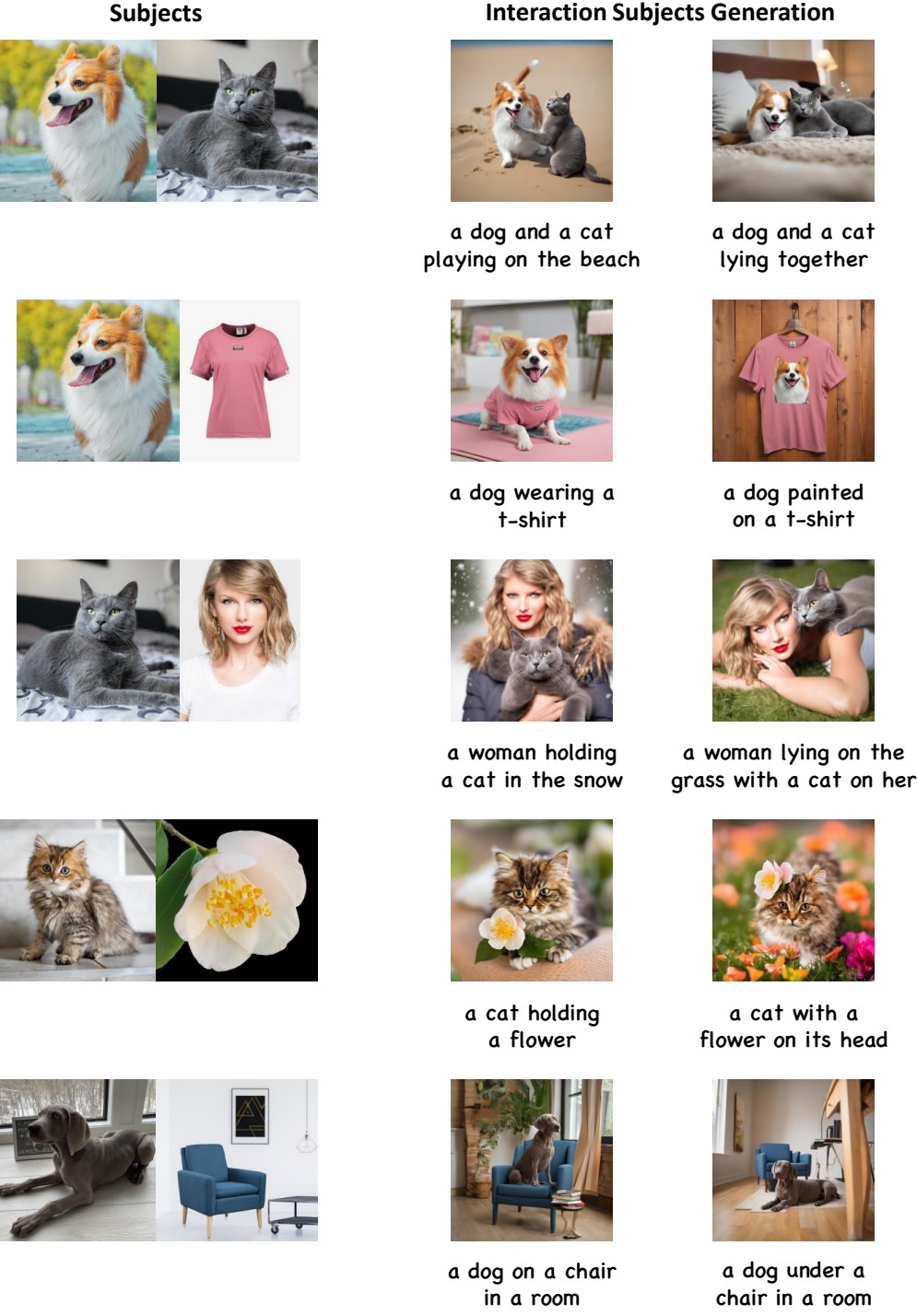

Figure 17: **Examples of prompts with complex interaction of multiple subjects.** MS-Diffusion can generate high-quality images following both the subjects and prompts.

**Subjects**                    **Results**

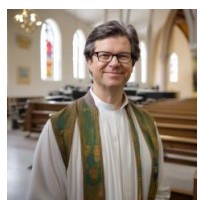

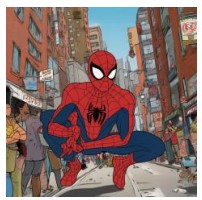

a man walking          a man standing
on the road            in the church

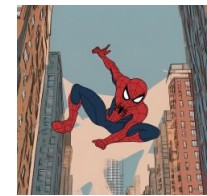

an anime man on        an anime man flying
a crowded street       between buildings

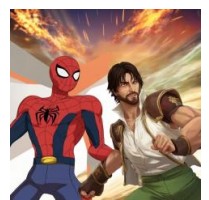

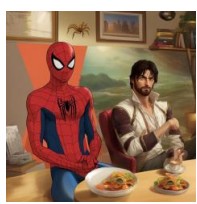

an anime man and       an anime man
an anime man           sitting besides an
fighting together      anime man

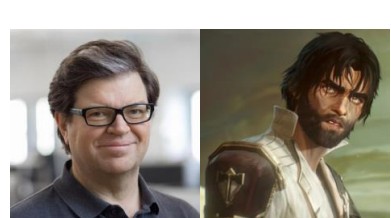

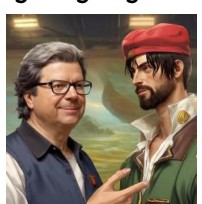

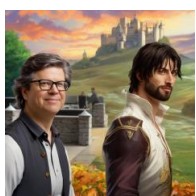

a man talking          a man and an anime
with an anime man      man with a castle in
wearing red hat        the background

Figure 18: **Personalized results of MS-Diffusion on human and anime subjects.** MS-Diffusion can generate high-quality images in both single-subject and multi-subject personalization for humans and anime.

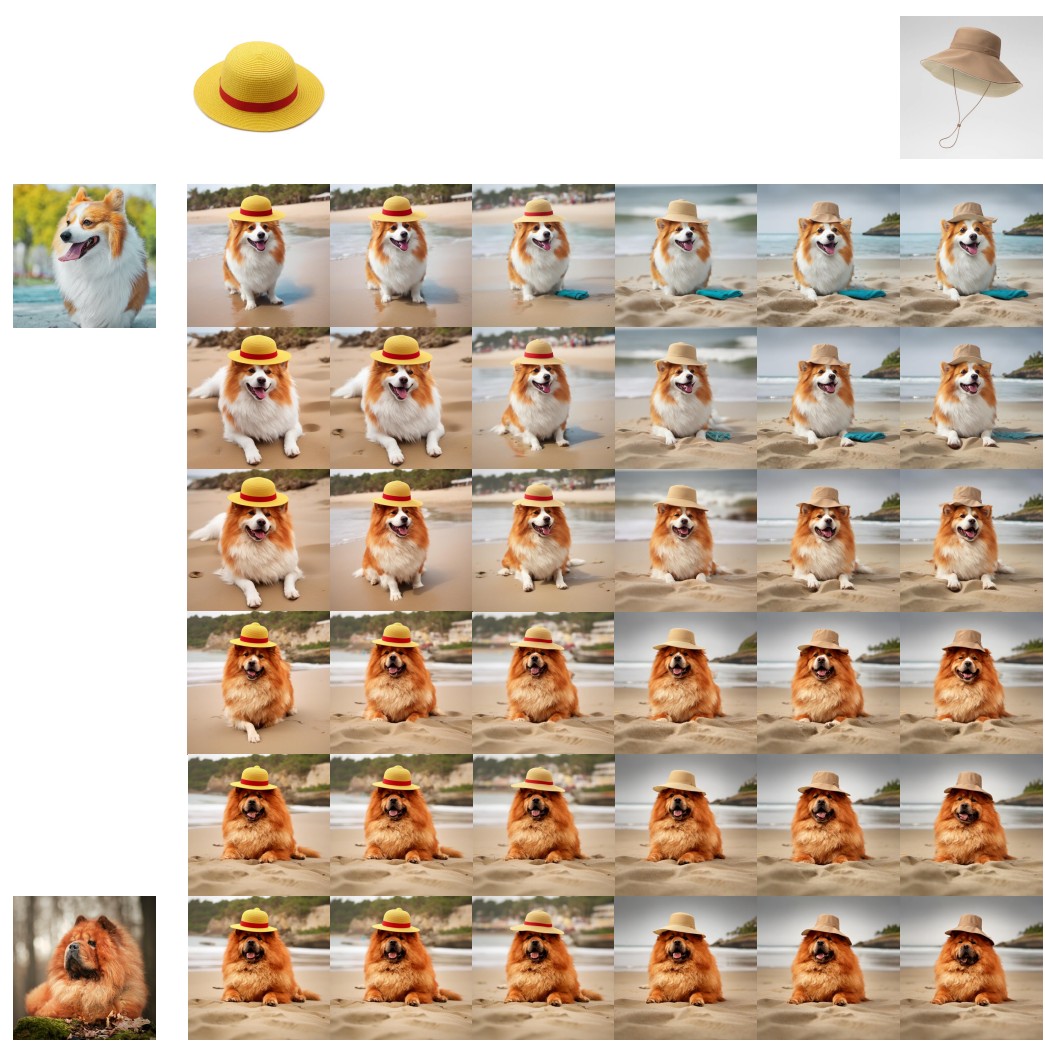

Figure 19: **Subjects interpolation in multi-subject generation.** We select two dogs and two hats to conduct linear interpolation with the text set to "a dog wearing a hat on the beach".

