# OpenReview forum: "MS-Diffusion: Multi-subject Zero-shot Image Personalization with Layout Guidance"
_ICLR.cc/2025/Conference — ICLR 2025 Poster_

### Official Review · Reviewer_CBea · 2024-10-28

**Soundness:** 3
**Presentation:** 3
**Contribution:** 3
**Rating:** 6
**Confidence:** 5

**Summary:**

This paper proposes multiple subjects (MS-Diffusion) framework, which consolidates the accommodation of multiple subjects, the incorporation of zero-shot learning capabilities, the provision of layout guidance, and the preservation of the foundational model's parameters.
MS-diffusion explicitly utilizes the layout information of the reference images to extract the information of multiple subjects separately to inject into the base model.

**Strengths:**

1.The authors alleviate the problem of combining natural objects in two thematic scenarios.
2.The framework is easy to think of and sensible.
3.The writing is clear and easy to understand.

**Weaknesses:**

1.Missing results for multiple topics (>2).
2.Missing results for comparison with paper [1].
3.The authors do not mention the Image encoder used.To the best of our knowledge, the Image encoder if it is a CLIP may lose the details of the themes, leading to the results in the last graph of Figure 4 and Figure 5.
4.From the CLIP-I scores in Table 2, it seems that text fidelity is not significantly improved.
5.How does it perform in scenes where the theme is people and anime?
[1]Cones 2: Customizable Image Synthesis with Multiple Subjects

**Questions:**

How does it perform in scenes where the theme is people and anime? Especially the problem of combining people with anime characters.

---

> ### Author Response · Authors · 2024-11-20
> **Response (1/1)**
>
> Q1: Missing results for multiple topics (>2).
>
> A1: Results for 3 subjects can be found in Fig.1 and Fig. 14.
>
> Q2: Missing results for comparison with Cones2.
>
> A2: We appreciate your valuable feedback. Comparison with Cones2 [1] has been provided in Fig. 8. Without tuning cost which exists in Cones2, MS-Diffusion still gets better detail preserving performance and outperforms Cones2 in text consistency.
>
> [1] Liu, Zhiheng, et al. "Cones 2: Customizable image synthesis with multiple subjects." *Proceedings of the 37th International Conference on Neural Information Processing Systems*. 2023. [paper link](https://arxiv.org/pdf/2305.19327)
>
> Q3: The authors do not mention the Image encoder used. To the best of our knowledge, the Image encoder if it is a CLIP may lose the details of the themes, leading to the results in the last graph of Figure 4 and Figure 5.
>
> A3: We agree with your thoughtful insights. We utilize CLIP-G as the image encoder in MS-Diffusion, which has inherent limitations in preserving detailed image features. To address this, we investigated the performance of MS-Diffusion and the baselines on uncommon subjects, as shown in Fig. 13 of the appendix. Though losing some details, MS-Diffusion significantly outperforms state-of-the-art methods, benefiting from the proposed grounding resampler.
>
> Q4: From the CLIP-I scores in Table 2, it seems that text fidelity is not significantly improved.
>
> A4: The image and text fidelity need to be balanced in the applications. As the model tends to memorize the given subject, most of the SOTAs struggle in controlling the generation using text prompts, resulting in low text fidelity. **MS-Diffusion achieves a substantial improvement in text fidelity without compromising image fidelity and even enhances its ability to preserve image details,** which is indicated in a high DINO score on single-subject benchmark and a high CLIP-T score on both benchmarks. As indicated in DreamBooth [2], DINO more accurately captures the similarity in details between the results and the labels, whereas CLIP-I may exhibit high scores in situations of background overfitting, resulting in a clear advantage for DINO, but a slight disadvantage for CLIP-I of MS-Diffusion.
>
> [2] Ruiz, Nataniel, et al. "Dreambooth: Fine tuning text-to-image diffusion models for subject-driven generation." *Proceedings of the IEEE/CVF conference on computer vision and pattern recognition*. 2023. [paper link](https://openaccess.thecvf.com/content/ICCV2023/papers/Van_Le_Anti-DreamBooth_Protecting_Users_from_Personalized_Text-to-image_Synthesis_ICCV_2023_paper.pdf)
>
> Q5: How does it perform in scenes where the theme is people and anime? Especially the problem of combining people with anime characters.
>
> A5: We have provided some results for human and anime subjects in Fig. 18 of the appendix. Despite the loss of some facial details caused by the CLIP image encoder, MS-Diffusion is still capable of generating high-quality results for these subjects.

---

> > ### Comment · Reviewer_CBea · 2024-11-23
> > **Refinement of questions**
> >
> > (1)Q1：I hope to see more quantitative results, not just hand-picked cases. How do scenarios perform if multiple themes make up a larger portion of the scenario?
> >
> > (2)Q5:It seems that the results are fair, are there some results with other methods？
> >
> > （3） I don't quite understand the explanation here. Are you expressing that MS-diffusion can alleviate the inherent limitations of CLIP?

---

> > > ### Author Response · Authors · 2024-11-23
> > > **Response to the refinement of questions**
> > >
> > > We appreciate your fast and thoughtful reply. Below is the response to the refinement of questions.
> > >
> > > Q6: I hope to see more quantitative results, not just hand-picked cases. How do scenarios perform if multiple themes make up a larger portion of the scenario?
> > >
> > > A6: We sincerely apologize for misunderstanding your request. As we clarified in Sec. B of the appendix, the proposed MS-Bench contains 4 types in combination with 3 subjects, including *living+living+living*, *object+object+object*, *living+object+scene*, and *upwearing+midwearing+downwearing*. We provide the quantitative results on these types of combinations below:
> > >
> > > | Method            | CLIP-I    | DINO      | M-DINO    | CLIP-T    |
> > > | ----------------- | --------- | --------- | --------- | --------- |
> > > | $\lambda$-ECLIPSE | 0.724     | 0.419     | 0.065     | 0.307     |
> > > | SSR-Encoder       | **0.728** | 0.420     | 0.069     | 0.288     |
> > > | **MS-Diffusion**  | 0.700     | **0.424** | **0.075** | **0.339** |
> > >
> > > **The results can reflect that the strength of MS-Diffusion in 3-subject combinations is more obvious than that of the whole benchmark.**
> > >
> > > [Anonymous link](https://ibb.co/xq27Chh) shows some cases when multiple subjects make up a larger portion of the scenario, where MS-Diffusion can also perform well in handling both the layout input and the personalization. However, as discussed in Sec. G of the appendix, MS-Diffusion is different from the layout-to-image model. The layout input is a weak condition compared to the reference images and the text. Although the generated results generally align with the input layout, they are rarely perfectly accurate in terms of size and position, often exhibiting slight deviations.
> > >
> > > Q7: It seems that the results for Q5 are fair, are there some results with other methods？
> > >
> > > A7: We provide a qualitative comparison in [anonymous link](https://ibb.co/M1dGYWx). In terms of both image and text fidelity, MS-Diffusion significantly outperforms the baselines.
> > >
> > > Q8: I don't quite understand the explanation here. Are you expressing that MS-diffusion can alleviate the inherent limitations of CLIP?
> > >
> > > A8: Your understanding is correct. In personalized text-to-image tasks, most methods, such as IP-Adapter, $\lambda$-ECLIPSE, and SSR-Encoder, adopt CLIP as the image encoder. As indicated in Fig. 13 and [anonymous link](https://ibb.co/M1dGYWx), MS-Diffusion significantly outperforms these methods in detail preservation capability. **We attribute this improvement to the proposed Grounding Resampler.** As shown in the ablation study of Tab. 3, processing CLIP features with this structure significantly enhances the DINO score.

---

> > > ### Author Response · Authors · 2024-11-25
> > > **Kindly reminder**
> > >
> > > Dear Reviewer CBea,
> > >
> > > Thank you for your continued engagement and for raising additional questions following our initial response. We have provided responses to address these concerns and would like to kindly follow up to ensure they align with your expectations.
> > >
> > > Please let us know if further clarifications are needed or if there are any remaining points you would like us to address. We greatly value your feedback and are eager to work towards resolving all concerns.
> > >
> > > Thank you again for your time and thoughtful input.
> > >
> > > Best regards,
> > > MS-Diffusion Authors

---

> ### Author Response · Authors · 2024-11-29
> **We kindly hope you might consider raising the score**
>
> Dear Reviewer CBea,
>
> Thank you very much for your thoughtful comments and detailed feedback, which have been incredibly valuable in improving our work. We have carefully addressed all the concerns raised and made substantial revisions to the manuscript:
>
> - **We add comparison with Cones2 in Fig. 8.** Without tuning cost which exists in Cones2, MS-Diffusion still gets better detail preserving performance and outperforms Cones2 in text consistency.
> - **We discuss the impact of the CLIP image encoder.** Though losing some details, MS-Diffusion significantly outperforms state-of-the-art methods, benefiting from the proposed Grounding Resampler. This approach enhances the model's detail-preserving capability without affecting the base model.
> - **We clarify the improvements of MS-Diffusion.** MS-Diffusion achieves a substantial improvement in text fidelity without compromising image fidelity and even enhances its ability to preserve image details, which is indicated in a high DINO score on single-subject benchmark and a high CLIP-T score on both benchmarks.
> - **We provide qualitative examples and comparisons on human and anime subjects in Fig. 18 and [anonymous link](https://ibb.co/M1dGYWx).** Despite the loss of some facial details caused by the CLIP image encoder, MS-Diffusion still demonstrates a significant advantage over baselines that also utilize CLIP. Moreover, as highlighted by Reviewer 7QyM, the performance of human subjects would improve if MS-Diffusion were to leverage facial datasets and a specialized face encoder.
> - **We provide results when multiple subjects make up a larger portion of the scenario in [anonymous link](https://ibb.co/xq27Chh)**. Since MS-Diffusion is not a layout-to-image model, the layout input is a weak condition compared to the reference images and the text. However, MS-Diffusion can also perform well in handling both the layout input and the personalization with slight deviations.
> - **We provide quantitative results on combinations with 3 subjects.** The results can reflect that the strength of MS-Diffusion in 3-subject combinations is more obvious than that of the whole benchmark.
>
> **If you find that our responses and the updated manuscript adequately address your concerns, we kindly hope you might consider raising the score. Your recognition of our efforts would mean a great deal to us.**
>
> Please let us know if there are any additional issues or improvements you would like us to address. We greatly appreciate your time and consideration.
>
> Wishing you a joyful and peaceful Thanksgiving!
>
> Best regards,
> MS-Diffusion Authors

---

> ### Author Response · Authors · 2024-12-03
> **Kindly reminder for the deadline of discussion**
>
> Dear Reviewer CBea,
>
> Thank you very much for initially assigning a positive score to our work. Over the course of this discussion period, we have been dedicated to addressing the various concerns raised, and we are grateful to have received recognition and positive feedback from the reviewers.
>
> As the deadline of discussion is in one day, we kindly hope you might consider further increasing the score if our responses have successfully resolved your concerns. This would be incredibly important for us, and we are committed to continuing our efforts to deliver high-quality research and impactful open-source projects in the future.
>
> Thank you again for your time and valuable insights.
>
> Best regards,
> MS-Diffusion Authors

---

### Official Review · Reviewer_Ebqr · 2024-11-03

**Soundness:** 3
**Presentation:** 3
**Contribution:** 2
**Rating:** 6
**Confidence:** 5

**Summary:**

Text-to-image generation models have advanced but face challenges in multi-subject scenarios. The MS-Diffusion framework is introduced to address these. It uses grounding tokens and feature resampler for detail, and layout guided cross-attention for location control. Experiments show it surpasses existing models in fidelity, promoting personalized text-to-image generation.

**Strengths:**

1.	The paper introduced the first layout-guided zero-shot image personalization with multiple subjects framework, which consolidates the accommodation of multiple subjects, the incorporation of zero-shot learning capabilities.

2.	The idea of decoupling and controlling the generation of the texture and position of the subject in the image is very reasonable.

3.	The paper is well written and easy to follow. The elaboration of the idea is very clear, and the framework of the structure diagram is also very easy to understand.

4.	The experiments are also very sufficient.

**Weaknesses:**

1.	The practice of controlling image generation through local cross attention is not innovative enough. It has been widely adopted in many existing layout-guided text-to-image generation methods.

2.	Judging from the quantitative and qualitative experimental results provided in the article, the improvement of image generation effect compared to existing methods is relatively limited.

3.	The extraction method of image features used by the grounding resampler proposed in the article has also been used in past papers, and its innovation is limited.

**Questions:**

1.	The grounding resampler proposed in the article also compresses image features. Why is it better than the image encoder used in existing methods? Can more qualitative analysis experiments be provided?

2.	The experiments in the article are based on Stable Diffusion XL. Now, newly emerging text-to-image diffusion models such as Flux inherently have better control over the generated content. Do these models not need an additionally trained structure to achieve satisfactory control?

---

> ### Author Response · Authors · 2024-11-20
> **Response (1/2)**
>
> Q1: The practice of controlling image generation through local cross-attention is not innovative enough. It has been widely adopted in many existing layout-guided text-to-image generation methods.
>
> A1: Some studies [1, 2] focus on mitigating conflicts in text cross-attentions. However, as we discussed in Sec 3.2, there are some limitations to the text cross-attention control. First, adjustments to text cross-attention can directly impact the control over text conditions. Second, text cross-attention does not directly dictate the areas of influence for image conditions; rather, it exerts an indirect influence on image conditions by shaping the image layout generated by the diffusion model. This indirect control may result in low performance and increased uncertainty. In comparison, to the best of our knowledge, MS-Diffusion is the **first** personalized text-to-image method employing **image attention masks**. **While addressing multi-object conflicts, MS-Diffusion ensures that text conditions remain unaffected, as evidenced by the significantly higher text adherence capability.**
>
> [1] Gu, Yuchao, et al. "Mix-of-show: Decentralized low-rank adaptation for multi-concept customization of diffusion models." *Advances in Neural Information Processing Systems* 36 (2024). [paper link](https://arxiv.org/pdf/2305.18292)
>
> [2] Ma, Jian, et al. "Subject-diffusion: Open domain personalized text-to-image generation without test-time fine-tuning." *ACM SIGGRAPH 2024 Conference Papers*. 2024. [paper link](https://arxiv.org/pdf/2307.11410)
>
> Q2: Judging from the quantitative and qualitative experimental results provided in the article, the improvement of the image generation effect compared to existing methods is relatively limited.
>
> A2: For qualitative comparison in Fig. 4 and Fig. 5, results of MS-Diffusion show high detail preservation capability and strong text control ability, especially L2, L4 in Fig. 4 and L2-4 in Fig. 5. For quantitative results, the image and text fidelity need to be balanced in the applications. As the model tends to memorize the given subject, most of the SOTAs struggle in controlling the generation using text prompts, resulting in low text fidelity. **MS-Diffusion achieves a substantial improvement in text fidelity without compromising image fidelity and even enhances its ability to preserve image details,** which is indicated by a high DINO score on single-subject benchmark and a high CLIP-T score on both benchmarks.
>
> On the multi-subject benchmark, the comparable image fidelity between MS-Diffusion and the baselines is primarily due to the overfitting issues in the baselines. To eliminate the inappropriate impact of overfitting on the metrics, as suggested by Reviewer MCZ6, we attempted to use a new metric, D&C (detect and compare) [3], to evaluate the performance of multi-subject personalization. This metric extracts each subject from the generated results and compares them with the input for evaluation. D&C can punish the subject neglect issue since there might be a missing subject that cannot be detected. **The significant advantage of MS-Diffusion under this metric demonstrates its powerful multi-subject personalization capability**:
>
> | Method            | D&C       | CLIP-T    |
> | ----------------- | --------- | --------- |
> | $\lambda$-ECLIPSE | 0.149     | 0.316     |
> | SSR-Encoder       | 0.119     | 0.303     |
> | **MS-Diffusion**  | **0.305** | **0.341** |
>
> [3] Jang, Sangwon, et al. "Identity Decoupling for Multi-Subject Personalization of Text-to-Image Models." *arXiv preprint arXiv:2404.04243* (2024). [paper link](https://arxiv.org/pdf/2404.04243)

---

> ### Author Response · Authors · 2024-11-20
> **Response (2/2)**
>
> Q3: The extraction method of image features used by the grounding resampler proposed in the article has also been used in past papers, and its innovation is limited.
>
> A3: Most of the layout-to-image methods [4, 5] have explored in injecting grounding information into the generation process of U-Net. However, to the best of our knowledge, MS-Diffusion is the **first** approach utilizing the grounding information in the image features projection. **As clarified in Sec 3.4, the purpose of this design is to indicate the resampler with semantic and positional prior, thus increasing the detail preserving ability.**
>
> [4] Wang, Xudong, et al. "Instancediffusion: Instance-level control for image generation." *Proceedings of the IEEE/CVF Conference on Computer Vision and Pattern Recognition*. 2024. [paper link](https://openaccess.thecvf.com/content/CVPR2024/papers/Wang_InstanceDiffusion_Instance-level_Control_for_Image_Generation_CVPR_2024_paper.pdf)
>
> [5] Li, Yuheng, et al. "Gligen: Open-set grounded text-to-image generation." *Proceedings of the IEEE/CVF Conference on Computer Vision and Pattern Recognition*. 2023. [paper link](http://openaccess.thecvf.com/content/CVPR2023/papers/Li_GLIGEN_Open-Set_Grounded_Text-to-Image_Generation_CVPR_2023_paper.pdf)
>
> Q4: The grounding resampler proposed in the article also compresses image features. Why is it better than the image encoder used in existing methods? Can more qualitative analysis experiments be provided?
>
> A4: Resampler uses learnable query tokens to extract image features. The proposed Grounding Resampler replaces the learnable query with grounding tokens, which means the resampler can extract the image features with grounding prior. As shown in Tab. 3 of the ablation study, using Grounding Resampler increases the DINO score (detail preserving). We also provide qualitative examples in Fig. 6, showing the difference in using Grounding Resampler or not, especially "dog" in L1 and L2, and "jacket" in L3.
>
> Q5: The experiments in the article are based on Stable Diffusion XL. Now, newly emerging text-to-image diffusion models such as Flux inherently have better control over the generated content. Do these models not need an additionally trained structure to achieve satisfactory control?
>
> A5: Similar to IP-Adapter for Flux, MS-Diffusion for Flux only requires injecting tokenized image features into the decoupled cross-attention, benefiting from the base model's freezing characteristics. Therefore, for Flux, MS-Diffusion only needs to train the image projector and cross-attentions, just like SDXL. We are working on open-sourcing MS-Diffusion, and in the future, we will consider extending it to the Flux model.

---

> ### Author Response · Authors · 2024-11-23
> **Kindly reminder**
>
> Dear Reviewer,
>
> I hope this message finds you well. We sincerely appreciate the time and effort you have dedicated to reviewing our submission. We have submitted our rebuttal and would like to follow up to inquire whether our responses have sufficiently addressed your concerns.
>
> Please let us know if you have any remaining questions or require additional clarification. We value your feedback and are eager to ensure our work meets the highest standards.
>
> Thank you again for your thoughtful insights and guidance.
>
> Best regards,
> MS-Diffusion Authors

---

> ### Comment · Reviewer_Ebqr · 2024-11-26
> **Thanks for the response.**
>
> Your reply is highly appreciated. I'd like to provide more thoughts and details regarding to the comments and author response.
>
> Specifically, as far as we know, previous grounded text-to-image generation methods [1],[2] have already utilized a similar masked attention mechanism to control the position of objects within images. Besides, previous work [3] has used learnable query for local feature learning and as the authors mentioned, using grounding information is a natural and common idea. In terms of experimental results, the authors introduced a new metric, but overall, the improvements in the experimental results seem not sufficiently convincing.
>
> [1] Xiao J, Lv H, Li L, et al. R&b: Region and boundary aware zero-shot grounded text-to-image generation[J]. arXiv preprint arXiv:2310.08872, 2023.
>
> [2] Phung Q, Ge S, Huang J B. Grounded text-to-image synthesis with attention refocusing[C]//Proceedings of the IEEE/CVF Conference on Computer Vision and Pattern Recognition. 2024: 7932-7942.
>
> [3] Arar M, Shamir A, Bermano A H. Learned queries for efficient local attention[C]//Proceedings of the IEEE/CVF Conference on Computer Vision and Pattern Recognition. 2022: 10841-10852.

---

> ### Author Response · Authors · 2024-11-26
> **Response to further thoughts and details**
>
> We appreciate your thoughtful feedback. Here, we would like to provide clarifications to address some potential misunderstandings.
>
> Q6: Previous grounded text-to-image generation methods have already utilized a similar masked attention mechanism to control the position of objects within images.
>
> A6: **As a personalized text-to-image model, MS-Diffusion can reference the input subject images to generate consistent results, which is fundamentally different from grounded text-to-image methods.** The primary role of the grounding information is to serve as guidance for resolving potential conflicts during multi-subject personalization, as indicated in Fig. 2 of the manuscript. Moreover, as clarified in Sec. 3.2 and A1 of the first response, MS-Diffusion is the **first** personalized text-to-image method to employ **image attention masks**. **This is only feasible in personalized models with image cross-attentions.** Previous grounded text-to-image methods utilized text attention masks, which could affect the base model's text-to-image generation capabilities. At the same time, text cross-attention does not directly dictate the areas of influence for image conditions; rather, it indirectly impacts image conditions by shaping the image layout generated by the diffusion model. **This indirect control may lead to lower performance and increased uncertainty.**
>
> Q7: Previous work has used learnable query for local feature learning and as the authors mentioned, using grounding information is a natural and common idea.
>
> A7: As discussed in A3 of the first response, the **main contribution** of the Grounding Resampler is to utilize the grounding information in the image feature projection rather than injecting it directly into the U-Net. **This approach enhances the model's detail-preserving capability without affecting the base model.** To the best of our knowledge, MS-Diffusion is the **first** method to propose such a module in personalized models, **making an important contribution to advancing the robust application of personalized models in real cases**, which is also highlighted by Reviewer 7QyM.
>
> Q8: In terms of experimental results, the authors introduced a new metric, but overall, the improvements in the experimental results seem not sufficiently convincing.
>
> A8: On the single-subject benchmark, MS-Diffusion demonstrates **significant advantages in detail preservation and text consistency**. On the multi-subject benchmark, as indicated in A2 of the first response, the comparable image fidelity between MS-Diffusion and the baselines is primarily due to overfitting issues in the baselines. Therefore, based on Reviewer MCZ6's suggestion, we introduced a new metric to mitigate the impact of overfitting. Experimental results in A2 further highlight **MS-Diffusion's superiority in multi-subject personalization**. The experiments evaluating the new metric were conducted based on the **official codebase**. **We are actively working on open-sourcing MS-Diffusion and commit to making its model weights directly accessible for use.**

---

> > ### Comment · Reviewer_Ebqr · 2024-11-27
> > **Thanks for the reply**
> >
> > Based on the discussion, I decided to raise my score to 6.

---

> > > ### Author Response · Authors · 2024-11-27
> > > **Thanks for your recognition**
> > >
> > > Dear Reviewer Ebqr,
> > >
> > > Thank you for raising the score! We truly appreciate your acknowledgment of our work and the constructive feedback you have provided. Your valuable insights enhance the quality of our research and motivate us to explore new frontiers in this field. We remain committed to improving this method and aim to deliver even more meaningful contributions in the future.
> > >
> > > Best regards,
> > > MS-Diffusion Authors

---

### Official Review · Reviewer_7QyM · 2024-11-03

**Soundness:** 3
**Presentation:** 2
**Contribution:** 2
**Rating:** 6
**Confidence:** 3

**Summary:**

The paper proposes a zero-shot method for multi-subject personalization of text-to-image models. This is done through a pre-training phase where an adapter is trained to adapt a diffusion model on additional conditions derived from input images. In particular, the authors use 3.6M videos to extract two views of the same subject. Each training sample comprises segmented entities, their bounding boxes and a ground truth frame including all entities.

The paper discusses mainly two design choices for their adapter, namely, a Grounded Resampler, and a Multi-Subject Cross Attention Layer. The grounded resampler is used to distill the subject-features through Preceiver like architecture. The authors propose to initialize the queries using entity word-embedding and bounding-box coordinates, to boost the localization of the sampling queries.

In addition, to avoid any information leakage between subjects, the authors use a Masked Cross attention where queries corresponding to a certain subject only attend the subject features.The proposed method also uses element-wise masking to avoid injected features from the reference images into background patches.

**Strengths:**

[1] The paper introduces a zero-shot multi-subject personalization method.

[2] There are interesting components to the proposed method like the data-collection from video and the Grounding Resampler Unit. However I feel like more in-depth analysis and ablation should be carried out to validate that design choices considered are indeed important.

**Weaknesses:**

[1] The paper is hard to follow and some parts need further clarification. In particular, The “Multi-subject Cross Attention” section is not clear and needs major revision.

Lack of ablation on critical parts of the methods:

[2] Using videos for Data collection (e.g, in which ref. Image is different from gt. image) is claimed to be a key contribution. However, there’s a lack of ablation on how much this method helps the authors.

[3] In the Grounding Resampler; how important is the initialization (words+bounding box) to the module localization capability ?

[4] Lacking qualitative samples of interesting use-cases such as - spatial relation (e.g, On-top) and object-interactions (e.g., two subjects playing tennis). The only interesting case I find in the paper is fitted to “virtual try-on” which could be a by-product of the training data.

**Questions:**

[1] L243-253: In the description of the cross-attention layers, in addition to the information injected from the text features, the authors show information that is injected from the reference image as well. In SD, the only features injected are those from the text. While injected image features through the cross-attention layers have been proposed (e.g, IP-Adapter), this wasn’t part of the original model. Please revise this section, and consider explicitly indicating the origin of the image features (f_i).

[2] Grounding Resampler: the authors initialize the queries with word-embeddings and bounding-box Sinusoidal/Fourier features. But in another part of the text, the authors claim these embeddings are learnable. I am confused, is it learnable because of the query project-matrices ? Or do you pass the entity-word-embedding along with Bounding-Box features to another Projection matrix ? Can you please clarify this point.   Please provide a step-by-step explanation of how the queries are initialized and then updated during training, clearly stating which components are learnable and which are fixed.

[3] Do the results in Figure 4 correspond to the MS-Diffusion model before or after fine tuning on DreamBench ? Please clearly label all results in the paper to indicate whether they are from zero-shot or fine-tuned models.

[4] The Multi Subject Cross Attention module has been used (e.g, in ELITE). To my understanding, MSDiffusion only extends this to multi-subject use cases. Can you please highlight the differences ?

[5] It would be nice to provide dataset statistics. In particular, what percentage of the data there are multiple-subjects, excluding trivial cases like a person and clothes. At least from the qualitative samples, it looks like the model performs very well on clothing entities and a single subject.

[6] Can you provide some information regarding the computational complexity of the method. In particular, it would be nice to report the additional overhead indeed by the method. Please report inference time and memory usage compared to baseline zero-shot models, for both single-subject and multi-subject generation tasks.

---

> ### Author Response · Authors · 2024-11-20
> **Response (1/2)**
>
> Q1: The "Multi-subject Cross-attention" section is not clear and needs revision.
>
> A1: We appreciate your valuable feedback. Multi-subject Cross-attention is developed to resolve the possible conflicts in multi-subject personalization, which is demonstrated in Fig. 2. We construct **attention masks** to ensure that each subject is represented within its designated area. Also, **dummy tokens and post attention masks** are introduced to ensure the control capability of texts. As suggested by Reviewer vy6x, we have tried to make this section clear in the revised manuscript. Please let us know if any detailed part needs further revision.
>
> Q2: There is a lack of ablation on the data collection pipeline.
>
> A2: Thanks for your thoughtful suggestion. As explained in Sec. A of the appendix, the data collection pipeline can help mitigate the model’s tendency to copy-and-paste. Based on our dataset, we make the ground truth and reference image the same to construct a non-video dataset for ablation. The results are reported below:
>
> single-subject:
>
> | Method            | CLIP-I | DINO  | CLIP-T |
> | ----------------- | ------ | ----- | ------ |
> | **MS-Diffusion**  | 0.792  | 0.671 | 0.321  |
> | w/o video dataset | 0.797  | 0.673 | 0.309  |
>
> multi-subject:
>
> | Method            | CLIP-I | DINO  | M-DINO | CLIP-T |
> | ----------------- | ------ | ----- | ------ | ------ |
> | **MS-Diffusion**  | 0.698  | 0.425 | 0.108  | 0.341  |
> | w/o video dataset | 0.685  | 0.420 | 0.099  | 0.327  |
>
> It is shown in the results that using the video dataset increases the text control capability of MS-Diffusion. **The image fidelity on single-subject benchmark slightly increases due to the overfitting. However, the image fidelity on multi-subject benchmark decreases since it struggles in complex combinations (like wearings) suffering from the copy-and-paste issue.**
>
> Q3: In the Grounding Resampler, how important is the initialization to the module localization capability? Please provide a step-by-step explanation of how the queries are initialized and then updated during training, clearly stating which components are learnable and which are fixed.
>
> A3: Grounding Resampler takes the grounding information as a prior, thus helping the resampler preserve the subject details. It also provides MS-Diffusion a layout control capability, which is indicated in Fig. 15 of the appendix. However, since MS-Diffusion is a personalized text-to-image model, we do not intend for it to rely on precise layout inputs like layout-to-image methods. Therefore, as we clarified in Sec. 3.4, the grounding tokens (fixed for each training sample) are randomly replaced by learnable tokens to enable MS-Diffusion to handle the situation when no layout prior is provided. A statement that might lead to misunderstanding has been corrected in the first paragraph of Sec. 3.4: Utilizing a set of learnable tokens, a **resampler** queries and distills pertinent information from the image features.
>
> Q4: Lacking qualitative samples of interesting use cases such as spatial relation and object interactions.
>
> A4: Benefiting from the grounding tokens and the proposed multi-subject cross-attention module, MS-Diffusion has advantages in handling spatial relations and object interactions, which is also highlighted by Reviewer MCZ6. We have provided more examples in the Fig. 17 of the appendix.
>
> Q5: Please explicitly indicate the origin of the image features in Sec. 3.1, which is newly introduced in SD.
>
> A5: Thank you for the valuable suggestion. As defined in "Stable Diffusion with Image Prompt", the image features are derived from a pre-trained image encoder and a trainable projector. We have revised this part from "In SD" to "In IP-Adapter" in the manuscript.
>
> Q6: Do the results in Figure 4 correspond to the MS-Diffusion model before or after fine-tuning on DreamBench? Please clearly label all results in the paper to indicate whether they are from zero-shot or fine-tuned models.
>
> A6: Thanks for your thoughtful comments. All qualitative results in the paper are generated by MS-Diffusion without fine-tuning on DreamBench. We have indicated that in Sec. 4.1 of the revised manuscript.

---

> ### Author Response · Authors · 2024-11-20
> **Response (2/2)**
>
> Q7: The Multi-Subject Cross-Attention module has been used in ELITE. To my understanding, MS-Diffusion only extends this to multi-subject use cases. Can you please highlight the differences?
>
> A7: The local mapping technique proposed by ELITE [1] is fundamentally different from the Multi-Subject Cross-Attention module. The purpose of local mapping is to mask out the subjects in the reference image, where segmentation masks are directly applied to the image features to **mitigate the influence of the background**. In contrast, the attention masks in the Multi-Subject Cross-Attention module are constructed from the input multi-object layout. These masks are designed to **specify the interaction relationships between query tokens and key tokens during cross-attention, thereby achieving the separation of multiple objects**.
>
> [1] Wei, Yuxiang, et al. "Elite: Encoding visual concepts into textual embeddings for customized text-to-image generation." *Proceedings of the IEEE/CVF International Conference on Computer Vision*. 2023. [paper link](https://openaccess.thecvf.com/content/ICCV2023/papers/Wei_ELITE_Encoding_Visual_Concepts_into_Textual_Embeddings_for_Customized_Text-to-Image_ICCV_2023_paper.pdf)
>
> Q8: It would be nice to provide dataset statistics. In particular, what percentage of the data there are multiple subjects, excluding trivial cases like a person and clothes.
>
> A8: As mentioned in Sec. A of the appendix, our dataset comprises 2.8M general scenario videos and 0.8M product demonstration videos. For the general scenario video dataset, it encompasses nearly all types of scenes, including landscapes, people, animals, and more. For the product demonstration video dataset, it features more explicit subject interactions. Though the products are primarily clothing items, the performance of clothing-related entities is hardly affected. Since the dataset is predominantly composed of general scenario videos, MS-Diffusion demonstrates excellent generalization capabilities across subjects in various scenarios.
>
> Q9: Can you provide some information regarding the computational complexity of the method. In particular, it would be nice to report the additional overhead indeed by the method. Please report inference time and memory usage compared to baseline zero-shot models, for both single-subject and multi-subject generation tasks.
>
> A9: We appreciate your valuable suggestion. The comparison of the inference time and memory usage compared to SDXL (base model) and IP-Adapter (baseline) is reported below:
>
> | Method           | Inference Time | GPU Memory |
> | ---------------- | -------------- | ---------- |
> | SDXL             | ~18s           | ~25.5G     |
> | IP-Adapter       | ~18s           | ~29.7G     |
> | **MS-Diffusion** | ~19s           | ~29.8G     |
>
> The experiments are run on an 80G NVIDIA A100 GPU, generating 5 images each run. **The additional overhead in MS-Diffusion only exists in the two proposed trainable modules, thus it has a slight impact on the entire pipeline.** The marginal increase in inference time compared to IP-Adapter is primarily due to the construction of attention masks.

---

> ### Author Response · Authors · 2024-11-23
> **Kindly reminder**
>
> Dear Reviewer,
>
> I hope this message finds you well. We sincerely appreciate the time and effort you have dedicated to reviewing our submission. We have submitted our rebuttal and would like to follow up to inquire whether our responses have sufficiently addressed your concerns.
>
> Please let us know if you have any remaining questions or require additional clarification. We value your feedback and are eager to ensure our work meets the highest standards.
>
> Thank you again for your thoughtful insights and guidance.
>
> Best regards,
> MS-Diffusion Authors

---

> ### Comment · Reviewer_7QyM · 2024-11-26
>
> Apologies for my delay in response, and thank you for the time you have put in replying to my inquiries.
>
> Here are two additional comments I have about the method:
>
> [1] After reviewing the responses I think the key-contribution of the paper is the usage of the video-dataset.
> Looking at the result, especially after reviewing Fig 17, I still find multi-subject interactions to be limited.
>
> [2] I find the re-sampler component to be interesting as well. Although such attention-based pooling has been proposed in different papers (QnA as suggested by Ebqr26, Perceiver-IO ), the re-sampler integration in the setting of Personalization of T2I models is interesting.
>
> [3] Although I haven't raised this issue in my review, but Fig 18 doesn't demonstrate good results on human and anime-characters. Specifically, sine Spiderman and Professor Yan LeCum are public figures, they are very likely to be in the training set of the base text-to-image model. Running SDXL on the prompt "Yan LeCun" yields images of similar people. So the gap needed to be learned by MS-Diffusion is not significant. Nonetheless, I think for human-centric use-cases, different data and possibly different Image Feature encoders may be needed.
>
>
> Given my above comment, I am raising my score to (6).

---

> > ### Author Response · Authors · 2024-11-26
> > **Thank you for the timely and constructive feedback**
> >
> > Dear Reviewer 7QyM,
> >
> > Thank you for the timely and constructive feedback! We sincerely appreciate your recognition of our work.
> >
> > As you mentioned, while MS-Diffusion has demonstrated a certain capability in handling multi-subject interactions compared to SOTAs, there is still room for improvement. We believe the main reason lies in the conditional control conflicts inherent in the text and image cross-attention architecture used by MS-Diffusion. Interactions are controlled by the text, whereas appearances are governed by the image. We are considering further improving MS-Diffusion's capability by explicitly addressing cases involving interactions.
> >
> > Additionally, we agree with your perspective on MS-Diffusion's handling of human subjects. As discussed with Reviewer CBea, MS-Diffusion is limited by the use of the CLIP image encoder, which results in some loss of facial detail during generation. Nevertheless, as shown in [anonymous link](https://ibb.co/M1dGYWx), MS-Diffusion still demonstrates a significant advantage over baselines that also utilize CLIP. We believe that if MS-Diffusion were to leverage facial datasets and a specialized face encoder, similar to human-personalized models, its performance on human subjects would improve significantly.
> >
> > Thank you once again for your valuable comments. We are committed to addressing these concerns and further enhancing the capabilities of MS-Diffusion.
> >
> > Best regards,
> > MS-Diffusion Authors

---

### Official Review · Reviewer_vy6x · 2024-11-04

**Soundness:** 3
**Presentation:** 3
**Contribution:** 2
**Rating:** 6
**Confidence:** 5

**Summary:**

This paper introduces MS-Diffusion, a zero-shot personalized image generation method for multiple subject entities. Given one or more reference subjects and the text prompt, the model generates high-quality target image based on the inputs accordingly, containing the subject following the desired text descriptions.
The main contribution of the paper is the proposal of integrating grounding information for multi-subject personalized generation, and a newly designed grounding resampler for incorporating the given grounding information.
By training a SDXL model with the encourage modules on a large-scale private dataset, the algorithm yield favorable results compared to previous designs.

**Strengths:**

1. High-quality multi-subject personalized generation.
 2. The novel design of the grounding resampler provides extraordinary performance boost compared to previous baselines.

**Weaknesses:**

• Lack of baseline comparisons and limited contributions
        ◦ The author claimed that they are the first to incorporate grounding information for multiple subject image personalization, however, plenty of existing works have proposed similar designs, for example Subject Diffusion, where the layout is also determined by bounding boxes as cross attention regularizations, while there are no related in-depth discussions.
        ◦ The comparison with previous design on the benchmark is also limited.

    • Limited performance boost
        ◦ In Table 2, the performance boost is rather trivial when comparing with SSR-Encoder on Multi-subject setting. Specifically, while SSR-encoder was trained based on SD1.5 and the proposed method is trained based on SDXL, the performance for is almost the same, except for a higher CLIP-T score.

**Questions:**

1. Comparing to method like Subject Diffusion and GLIGEN, what makes proposed layout control novel?
    2. In figure 2, the author proposed several limitation cases identified from previous methods, however, they are not further discussed or addressed in the following paragraphs, for example, how is subject overcontrol defined? It usually better to have some specific discussion to recall and address the problems mentioned in the introduction.
    3. On the multi-subject generation benchmark, the performance boost compared to the baseline SSR-encoder remains minimal, despite the proposed method being trained on a larger and more advanced base model (SDXL versus SD1.5). This outcome appears to serve as a counterexample, challenging the effectiveness of the proposed design.
    4. The seems to produce a higher CLIP-T scores (better text following) than previous methods, however, this performance boost might also come from the usage of SDXL where multiple text encoder is designed specifically for increase text adherence. Please justify this boost comes from the proposed model design rather than the base model.

---

> ### Author Response · Authors · 2024-11-20
> **Response (1/2)**
>
> Q1: Lack of baseline comparisons and limited contributions. Compared to methods like Subject Diffusion and GLIGEN, what makes the proposed layout control novel?
>
> A1: Most of the relevant methods have explored in injecting grounding information into the generation process of U-Net. GLIGEN [1] encodes the grounding information into self-attention tokens, and Subject Diffusion [2] handles the grounding information in a plug-in adapter. Though introducing grounding priors, they require the text-to-image backbone to adapt to the injected inputs, which can affect the text-to-image capabilities of the pre-trained model. It is noteworthy that Subject Diffusion is developed on a similar task with MS-Diffusion. While being effective in multi-subject personalization, it suffers from a copy-and-paste issue. This leads to the model struggling in subject control, indicated by the low text CLIP-T of Subject-Diffusion on DreamBench:
>
> | Methods           | CLIP-I    | DINO      | CLIP-T    |
> | ----------------- | --------- | --------- | --------- |
> | Subject-Diffusion | 0.787     | **0.711** | 0.293     |
> | **MS-Diffusion**  | **0.792** | 0.671     | **0.321** |
>
> A further qualitative comparison is provided in [anonymous link](https://ibb.co/ZKvh6Zk). Since Subject Diffusion has not opened the checkpoint, we derive the results from the paper [2].
>
> To the best of our knowledge, MS-Diffusion is the **first** approach utilizing the grounding information in the image features projection. **As clarified in Sec 3.4, the purpose of this design is to indicate the resampler with semantic and positional priors, thus increasing the detail preserving ability.** Moreover, this design only adds grounding information to the image tokens of cross-attentions, having no influence on the original text-to-image backbone.
>
> [1] Li, Yuheng, et al. "Gligen: Open-set grounded text-to-image generation." *Proceedings of the IEEE/CVF Conference on Computer Vision and Pattern Recognition*. 2023. [paper link](http://openaccess.thecvf.com/content/CVPR2023/papers/Li_GLIGEN_Open-Set_Grounded_Text-to-Image_Generation_CVPR_2023_paper.pdf)
>
> [2] Ma, Jian, et al. "Subject-diffusion: Open domain personalized text-to-image generation without test-time fine-tuning." *ACM SIGGRAPH 2024 Conference Papers*. 2024. [paper link](https://arxiv.org/pdf/2307.11410)
>
> Q2: Several limitation cases should be further discussed or addressed.
>
> A2: Thank you for the thoughtful feedback. We have discussed the limitation cases in Section 3.5. All these cases are handled by the Multi-subject Cross-attention. For subject neglect and subject conflict, MS-Diffusion makes each subject represent in the according area. The design of dummy image tokens and post attention mask ensure that the area without any subject can be controlled by the text prompt, preventing subject overcontrol.

---

> ### Author Response · Authors · 2024-11-20
> **Response (2/2)**
>
> Q3: On the multi-subject generation benchmark, the performance boost compared to the baseline SSR-encoder remains minimal, despite the proposed method being trained on a larger and more advanced base model (SDXL versus SD1.5). A higher CLIP-T scores might also come from the usage of SDXL.
>
> A3: The image and text fidelity need to be balanced in the applications. As the model tends to memorize the given subject, most of the SOTAs struggle to control the generation using text prompts, resulting in low text fidelity. **MS-Diffusion achieves a substantial improvement in text fidelity without compromising image fidelity and even enhances its ability to preserve image details,** which is indicated by a high DINO score on single-subject benchmark and a high CLIP-T score on both benchmarks.
>
> Though SDXL demonstrates stronger text adherence capabilities compared to SD1.5, the differences observed in our experiments on simple prompts from DreamBench and MS-Bench are subtle. For each prompt, we generated five images using both SDXL and SD1.5 and calculated the CLIP-T score. The results in the table show that the base model has a much smaller impact on CLIP-T compared to the significant advantage of MS-Diffusion in CLIP-T, highlighting the substantial improvement in MS-Diffusion's text control capabilities. **Notably, MS-Diffusion has almost no impact on the text adherence capability of SDXL, whereas the SSR-Encoder causes a degradation of this capability in SD1.5**:
>
> | Method                  | CLIP-T |
> | ----------------------- | ------ |
> | SD1.5                   | 0.338  |
> | SDXL                    | 0.348  |
> | SSR-Encoder (SD1.5)     | 0.303  |
> | **MS-Diffusion (SDXL)** | 0.341  |
>
> On the multi-subject benchmark, the comparable image fidelity between MS-Diffusion and the baselines is primarily due to the overfitting issues in the baselines. To eliminate the inappropriate impact of overfitting on the metrics, as suggested by Reviewer MCZ6, we attempted to use a new metric, D&C (detect and compare) [3], to evaluate the performance of multi-subject personalization. This metric extracts each subject from the generated results and compares them with the input for evaluation. D&C can punish the subject neglect issue since there might be a missing subject that cannot be detected. **The significant advantage of MS-Diffusion under this metric demonstrates its powerful multi-subject personalization capability**:
>
> | Method            | D&C       | CLIP-T    |
> | ----------------- | --------- | --------- |
> | $\lambda$-ECLIPSE | 0.149     | 0.316     |
> | SSR-Encoder       | 0.119     | 0.303     |
> | **MS-Diffusion**  | **0.305** | **0.341** |
>
> [3] Jang, Sangwon, et al. "Identity Decoupling for Multi-Subject Personalization of Text-to-Image Models." *arXiv preprint arXiv:2404.04243* (2024). [paper link](https://arxiv.org/pdf/2404.04243)

---

> ### Author Response · Authors · 2024-11-23
> **Kindly reminder**
>
> Dear Reviewer,
>
> I hope this message finds you well. We sincerely appreciate the time and effort you have dedicated to reviewing our submission. We have submitted our rebuttal and would like to follow up to inquire whether our responses have sufficiently addressed your concerns.
>
> Please let us know if you have any remaining questions or require additional clarification. We value your feedback and are eager to ensure our work meets the highest standards.
>
> Thank you again for your thoughtful insights and guidance.
>
> Best regards,
> MS-Diffusion Authors

---

> ### Author Response · Authors · 2024-11-27
> **Kindly reminder for the deadline of manuscript revision**
>
> Dear Reviewer vy6x,
>
> As the deadline for manuscript revisions is less than a day away, I wanted to kindly follow up regarding the concerns you raised earlier. We have already provided detailed responses to address your feedback, but we have not yet received any further comments or suggestions.
>
> If there are any remaining points or clarifications needed, please feel free to let us know. We greatly value your insights and are eager to ensure the final manuscript meets your expectations.
>
> Thank you for your time and thoughtful consideration.
>
> Best regards,
> MS-Diffusion Authors

---

> > ### Comment · Reviewer_vy6x · 2024-11-29
> >
> > Thanks for the author's feedback. The response partially addressed my concern. I am happy to raise the rating.

---

> > > ### Author Response · Authors · 2024-11-29
> > > **Thank you for raising the rating**
> > >
> > > Dear Reviewer vy6x,
> > >
> > > Thank you very much for your kind words and for raising the rating. We sincerely appreciate your thoughtful feedback and are glad that our responses have addressed some of your concerns. If there are any remaining points that you feel need further discussion or clarification, we would be more than happy to engage and make the necessary improvements.
> > >
> > > We remain committed to enhancing our work and ensuring it meets the highest standards. Thank you again for your time and invaluable input.
> > >
> > > Wishing you a joyful and peaceful Thanksgiving!
> > >
> > > Best regards,
> > > MS-Diffusion Authors

---

### Official Review · Reviewer_MCZ6 · 2024-11-07

**Soundness:** 3
**Presentation:** 3
**Contribution:** 3
**Rating:** 6
**Confidence:** 3

**Summary:**

The paper introduces the MS-Diffusion framework, a layout-guided zero-shot image personalization approach for multi-subject scenarios. The author features a grounding resampler to enhance subject fidelity with semantic and positional priors and a novel cross-attention mechanism to ensure that each subject is represented in specific areas and facilitating the integration of multi-subject data while mitigating conflicts between text and image subject control.

**Strengths:**

- The paper is well-written and well-organized, presenting a clear and compelling motivation for the study.
- This method is the first to introduce layout-guided zero-shot image personalization for multi-subject scenarios.
- The paper showcases impressive qualitative results, particularly in layout control capabilities across both single- and multi-subject personalization, as well as in handling prompts with complex interactions among multiple subjects.

**Weaknesses:**

- In Section 2.2, second paragraph, the sentence "Though past research in this field has significantly enhanced the ability to reference single subjects, few zero-shot multi-subject personalized models" is a bit unclear. I suggest rephrasing this sentence for clarity.

- In Section 3.1, as well as in the rest of the paper, it appears that the transpose notation (T) is missing for K in the equations of calculating attention maps.

**Questions:**

- Could the authors provide a bit more detail on how the M-DINO metric is calculated and how it measures multi-subject fidelity?

- I came across a recent paper [1] on a similar task and recommend testing its evaluation metric on a few cases to see if it could be helpful for assessing multi-subject fidelity.

- The qualitative results presented are impressive, particularly in handling prompts with complex interactions among multiple subjects. Could the authors provide additional examples demonstrating these capabilities, as well as discuss any current limitations?

[1] Identity Decoupling for Multi-Subject Personalization of Text-to-Image Models (https://arxiv.org/pdf/2404.04243)

---

> ### Author Response · Authors · 2024-11-20
> **Response (1/1)**
>
> Q1: The sentence "Though past research in this field has significantly enhanced the ability to reference single subjects, few zero-shot multi-subject personalized models" in Sec 2.2 is unclear.
>
> A1: Thanks for your valuable suggestion. We have rephrased this sentence: **Though past research in this field has significantly enhanced the ability to reference single subjects, few studies have explored zero-shot multi-subject personalized models.**
>
> Q2: The transpose notation (T) is missing for K in the equations of attention.
>
> A2: We appreciate your careful reading. The transpose annotations have been added in the revised manuscript.
>
> Q3: Details on how the M-DINO metric is calculated and how it measures multi-subject fidelity.
>
> A3: We have provided the definition of M-DINO in Sec. 4.1 (Evaluation metrics). As indicated in Case A of Figure 2, a subject neglect issue is suffered by baselines, making the image fidelity of the missing subject obviously low but the others are normal. Therefore, we utilize the product of multi-subject DINO score to detect the subject neglect.
>
> Q4: Suggestion on using a new metric designed for multi-subject personalization.
>
> A4: We appreciate your helpful suggestions. Considering the subject neglect issue and possible failure in detecting subjects in the results, we calculate the similarity between the whole result images and input subjects, respectively. We have applied the mentioned metric, D&C [1], in our experiments. D&C can punish the subject neglect issue since there might be a missing subject that cannot be detected. The results are reported below:
>
> | Method            | D&C       | CLIP-T    |
> | ----------------- | --------- | --------- |
> | $\lambda$-ECLIPSE | 0.149     | 0.316     |
> | SSR-Encoder       | 0.119     | 0.303     |
> | **MS-Diffusion**  | **0.305** | **0.341** |
>
> **This metric proves to be extremely beneficial for MS-Diffusion, highlighting our strengths more prominently.**
>
> [1] Jang, Sangwon, et al. "Identity Decoupling for Multi-Subject Personalization of Text-to-Image Models." *arXiv preprint arXiv:2404.04243* (2024). [paper link](https://arxiv.org/pdf/2404.04243)
>
> Q5: The qualitative results presented are impressive, particularly in handling prompts with complex interactions among multiple subjects. Could the authors provide additional examples demonstrating these capabilities, as well as discuss any current limitations?
>
> A5: Thanks for your approvement. Benefiting from the multi-subject dataset and the proposed multi-subject cross-attention module, MS-Diffusion has advantages in handling complex interactions. We have provided more examples in Fig. 17 of the appendix. In interaction generation, a current limitation of MS-Diffusion lies in its reliance on preset layouts. For complex and detailed interactions, users may need to adjust the layout multiple times to achieve high-quality generation results. The pseudo layout guidance mentioned in Sec. M can alleviate this issue; however, the text attention maps do not fully correspond accurately to the layout during generation.

---

> > ### Comment · Reviewer_MCZ6 · 2024-11-23
> > **Response**
> >
> > I appreciate the authors' response. The additional results and clarifications provided have addressed my concerns, and I am pleased to raise my rating.

---

> > > ### Author Response · Authors · 2024-11-24
> > > **We sincerely appreciate your recognition**
> > >
> > > Dear Reviewer MCZ6，
> > >
> > > Thank you for raising the score! We sincerely appreciate your recognition of our work and your valuable feedback. Your thoughtful insights help improve our research and inspire us to push the boundaries of this field. We will continue refining this method and hope to contribute even more impactful work in the future.
> > >
> > > Best regards,
> > > MS-Diffusion Authors

---

> ### Author Response · Authors · 2024-11-23
> **Kindly reminder**
>
> Dear Reviewer,
>
> We hope this message finds you well. We sincerely appreciate the time and effort you have dedicated to reviewing our submission. We have submitted our rebuttal and would like to follow up to inquire whether our responses have sufficiently addressed your concerns.
>
> Please let us know if you have any remaining questions or require additional clarification. We value your feedback and are eager to ensure our work meets the highest standards.
>
> Thank you again for your thoughtful insights and guidance.
>
> Best regards,
> MS-Diffusion Authors

---

### Author Response · Authors · 2024-11-20
**Global Response**

We thank the reviewers for their thoughtful reviews. We are encouraged with positive feedback on the **novel method** (MCZ6, vy6x, Ebqr), **reasonable and interesting module design** (7QyM, Ebqr), **impressive and sufficient experiment results** (MCZ6, vy6x, Ebqr), **effective problem solution** (CBea), and **well-structured manuscript and presentation** (MCZ6, Ebqr, CBea).

In the revised manuscript, we update some clarifications according to the suggestions from all reviewers. To better demonstrate the ability of our model, we add more qualitative examples in the appendix:

- Figure 8 (updated): Qualitative comparison of Mix-of-Show, Cones2, and MS-Diffusion
- Figure 17 (updated): Examples of prompts with complex interaction of multiple subjects
- Figure 18 (new): Personalized results of MS-Diffusion on human and anime subjects

Moreover, there are some points we want to highlight:

- **The novelty of proposed modules.** The key modules of MS-Diffusion are Grounding Resampler and Multi-subject Cross-attention, which are both different from the existing research.
  - Grounding Resampler: Most of the layout-to-image methods have explored in injecting grounding information into the generation process of U-Net. However, to the best of our knowledge, MS-Diffusion is the **first** approach utilizing the grounding information in the image features projection. **As clarified in Sec 3.4, the purpose of this design is to indicate the resampler with semantic and positional prior, thus increasing the detail preserving ability.**
  - Multi-subject Cross-attention: Some studies focus on mitigating conflicts in text cross-attentions. However, as we discussed in Sec 3.2, there are some limitations for the text cross-attention editing, resulting in low performance and increased uncertainty. In comparison, to the best of our knowledge, MS-Diffusion is the **first** personalized text-to-image method employing **image attention masks**. **While addressing multi-object conflicts, MS-Diffusion ensures that text conditions remain unaffected, as evidenced by the significantly higher text adherence capability.**
- **The improvement of image and text fidelity.** These metrics need to be balanced in the applications. As the model tends to memorize the given subject, most of the SOTAs struggle in controlling the generation using text prompts, resulting in low text fidelity. **MS-Diffusion achieves a substantial improvement in text fidelity without compromising image fidelity and even enhances its ability to preserve image details.**

- **We are advocates of open research and strive to make everything publicly accessible. We are endeavoring to make the model available to the public before paper publication.**

Detailed responses to each reviewer are provided below. All revisions of the manuscript are marked as blue. Please let us know if any further clarification or discussion is required.

---

### Author Response · Authors · 2024-11-27
**Kindly reminder for the deadline of manuscript revision**

Dear Reviewers,

As the deadline for manuscript revisions is less than a day away, we wanted to kindly follow up to confirm if there are any additional points or updates you would like us to address based on your reviews.

We have already made revisions to the manuscript according to your feedback, and we greatly value your input to ensure the final version meets your expectations. Please feel free to let us know if there are any remaining suggestions.

Thank you once again for your time and thoughtful comments.

Best regards,
MS-Diffusion Authors

---

### Meta-Review · Area_Chair_3m2P · 2024-12-21

**Metareview:**

This paper introduces MS-Diffusion, a new approach for zero-shot image personalization that incorporates multiple subjects and uses layout guidance. The proposed grounding resampler and multi-subject cross-attention mechanism aim to enhance image generation tasks by focusing on particular areas and subjects within an image. The reported results suggest that the method can achieve qualitative and quantitative improvements over existing techniques.

The paper's main strengths lie in its novel contributions to the field of zero-shot image personalization. The introduction of the grounding resampler and cross-attention mechanism is innovative and has the potential to push forward the capabilities of image generation models. Additionally, the paper is well-structured and effectively communicates the proposed methodologies and their potential impact.

Reviewers have pointed out concerns regarding the level of innovation in the use of local cross-attention for controlling image generation. Questions about the comparative performance of the grounding resampler to existing methods were raised, as well as the need for more comprehensive quantitative evaluations. There were also criticisms about the loss of detail when using the CLIP image encoder.

The paper provides valuable contributions to zero-shot image personalization and has effectively addressed the main concerns raised during the review process. The remaining points of criticism are not critical enough to override the decision to accept, given the overall strength of the paper and the positive changes made in response to the review feedback.

**Additional Comments On Reviewer Discussion:**

During the discussion phase, the reviewers brought forth several points of concern:
- Reviewer MCZ6 questioned the novelty in the use of local cross-attention.
- Reviewer vy6x emphasized the need for a stronger comparative analysis.
- Reviewers 7QyM and Ebqr pointed out issues with the CLIP encoder and detail preservation.
- Reviewer CBea called for a more thorough evaluation on diverse character types.

The authors addressed these points in their rebuttal by elaborating on the uniqueness of their approach, providing more comparative examples, detailing the grounding resampler's advantages, introducing a new evaluation metric, and including results for both human and anime characters. Most concerns were resolved except for Reviewer CBea's request for additional quantitative results.

---

### Decision · Program_Chairs · 2025-01-22

Accept (Poster)